# Airborne quantification of Angolan offshore oil and gas methane emissions

Alina Fiehn<sup>1</sup>, Maximilian Eckl<sup>1</sup>, Magdalena Pühl<sup>1</sup>, Tiziana Bräuer<sup>1</sup>, Klaus-Dirk Gottschaldt<sup>1</sup>, Heinfried Aufmhoff<sup>1</sup>, Lisa Eirenschmalz<sup>1,2</sup>, Gregor Neumann<sup>1</sup>, Felicitas Sakellariou<sup>1</sup>, Daniel Sauer<sup>1</sup>, Robert Baumann<sup>1</sup>, Guilherme De Aguiar Ventura<sup>3</sup>, Winne Nayole Cadete<sup>3</sup>, Dario Luciano Zua<sup>3</sup>, Manuel Xavier<sup>4</sup>, Paulo Correia<sup>4</sup>, Anke Roiger<sup>1</sup>

- <sup>1</sup> Deutsches Zentrum für Luft- und Raumfahrt, Institut für Physik der Atmosphäre, Oberpfaffenhofen, Germany
- <sup>2</sup> Deutsches Zentrum für Luft- und Raumfahrt, Flugexperimente, Oberpfaffenhofen, Germany
- <sup>3</sup> Agência Nacional de Petróleo, Gás e Biocombustíveis, Luanda, Angola
- 15 <sup>4</sup> Ministério dos Recursos Minerais, Petróleo e Gás, Luanda, Angola

Correspondence to: Alina Fiehn (alina.fiehn@dlr.de)

Abstract. In September 2022, the METHANE-To-Go Africa (MTGA) scientific aircraft campaign, part of UNEP's International Methane Emissions Observatory (IMEO) Methane Science Studies, conducted the first methane (CH<sub>4</sub>) emissions measurements from the offshore oil and gas sector in West Africa. Using aircraft-based mass balance methods, we quantified trace gas emissions from all 57 Angolan offshore facilities, estimating total sector emissions and assessing 30 individual sites and 10 facility groups providing the first independent dataset for this previously unstudied region. Emissions were generally consistent across repeated measurements, but two facilities showed intermittent high emissions of 10 and 4 t h<sup>-1</sup>, significantly influencing total emissions. Older, shallow-water platforms emitted more than newer, deep-water floating production facilities. These findings suggest, that production volume is a poor proxy for CH<sub>4</sub> emissions, while age and maintenance status are more reliable indicators. Due to operational variability, regular monitoring remains essential, particularly for older facilities. We estimate total CH<sub>4</sub> emissions at  $16.9 \pm 5.3$  t h<sup>-1</sup>, only 20-22% of EDGAR and CAMS inventory estimates, but over twice the operator-reported values. Additional trace gas measurements (CO<sub>2</sub>, CO, C<sub>2</sub>H<sub>6</sub>, SO<sub>2</sub>, NOy, and aerosols) suggest CH4 stems primarily from fugitive emissions and venting, not combustion. The calculated carbon intensity of Angolan offshore oil and gas is  $3.4 \pm 0.8$  g CO<sub>2</sub>eq MJ<sup>-1</sup>, with nearly equal contributions from CH<sub>4</sub> and CO<sub>2</sub>. Notably, shallow-water platforms are CH<sub>4</sub>-dominated, while deep-water facilities emit mostly CO<sub>2</sub>. These results improve our understanding of greenhouse gas emissions from offshore production in a key oil- and gas-producing region.

#### 1 Introduction

Atmospheric methane (CH<sub>4</sub>) concentrations have more than doubled since the beginning of the industrial age, making it the second most significant anthropogenic greenhouse gas after carbon dioxide (CO<sub>2</sub>).

The 2015 Paris Agreement of the United Nations Framework Convention on Climate Change (UNFCCC) aims at limiting the global warming to below 1.5°C (UNFCCC, 2015). With a global warming potential 80–83 times that of CO<sub>2</sub> over a 20-year time horizon contributes 16% to the effective radiative forcing of well-mixed greenhouse gases since 1750 (IPCC, 2023). Considering its short lifetime of around a decade, CH<sub>4</sub> presents a high potential for mitigation strategies aimed at achieving the UNFCCC Paris Agreement's goal to mitigate climate warming (Nisbet et al., 2019). Recently, Angola signed up to the Global Methane Pledge, aiming to cut global CH<sub>4</sub> emissions by at least 30% from 2020 levels by 2030 (European Commission, United States of America, 2021).

The O&G sectors have been estimated to account for 22% (18–27%) of global anthropogenic CH<sub>4</sub> emissions (bottom-up in 2017; Saunois et al., 2020). Approximately 30% of global O&G production occurs offshore (IEA World Energy Outlook).

- This includes significant contributions from major offshore producing regions, such as the Gulf of Mexico, the North Sea, Brazil, West Africa, and Southeast Asia. CH<sub>4</sub> is emitted during routine operations on offshore O&G platforms for safety and operational reasons (e.g., shutdown or start-up of equipment during production) by either controlled venting or flaring. In the latter case, CO<sub>2</sub> is released simultaneously, with the CH<sub>4</sub> to CO<sub>2</sub> emission ratio dependent on the flaring efficiency. Another source of methane emission are unintended leaks on O&G installations.
- Recent research highlights significant discrepancies between bottom-up inventory estimates and measurement-based assessments of greenhouse gas emissions from oil and gas production. Several studies indicate that bottom-up methane inventories often underestimate emissions from the O&G industry (Schwietzke et al., 2016; Saunois et al., 2020; MacKay et al., 2021; Gorchov Negron et al., 2023). Unintended leaks as well as blow-outs can significantly contribute to CH<sub>4</sub> emissions (Lyon et al., 2015; Conley et al., 2016; Zavala-Araiza et al., 2017; Lee et al., 2018; Pandey et al., 2019; Varon et al., 2019).
- Similar underestimations have also been documented for CO<sub>2</sub>. For example, aircraft and satellite observations over the Canadian oil sands revealed that CO<sub>2</sub> emissions are consistently underestimated in bottom-up inventories, as shown by Liggio et al. (2019) and Wren et al. (2023). Likewise, Gorchov Negron et al. (2024) used atmospheric observations to assess the carbon intensity of U.S. offshore oil and gas production, concluding that measurement-based estimates frequently exceed industry-reported values.
- Top-down emission estimates from direct measurements close to sources can help to independently validate bottom-up estimates in inventory data. Better understanding, monitoring, and verification of CH<sub>4</sub> emissions associated with O&G operations are crucial parts of the Global Methane Pledge (European Commission and United States of America, 2021). Emissions from offshore O&G facilities are especially uncertain. Observations are sparse, partly because offshore facilities are less accessible, but also because the satellite detection and quantification of offshore methane plumes are highly challenging

due to the low albedo of the ocean surface in the relevant wavelengths. Therefore, only the biggest plumes are detected and only during favorable weather conditions. The smallest plume detected so far by satellite in offshore Angola is 0.8 t h<sup>-1</sup> (UNEP, 2024). Airborne in-situ mass balance is currently the most reliable technique for assessing offshore methane emissions, because it has a low detection limit, good spatial coverage and can also be conducted under cloudy conditions.

Africa is a significant contributor to the global oil and gas (O&G) industry, accounting for approximately 8% of global crude oil production and 6% of global natural gas production in 2022 (IEA, 2023). Nigeria and Angola are the continent's top producers, together contributing nearly 50% of Africa's oil output. In particular, Angola ranks among the top 20 globally, producing approximately 1.1 million barrels of oil per day in 2022 (OPEC, 2023). Most of Angola's production comes from offshore deep-water fields, which are technically complex and energy-intensive to operate, but have newer infrastructure than the shallow-water fields.

More particular, the country's offshore oil production is split between older shallow-water platforms closer to the coast and newer deep-water and ultradeep-water fields operated by tethered Floating Production Storage and Offloading (FPSO) vessels that can serve several oil fields at once and therefore have higher production volumes than the shallow-water platforms. Much of the produced natural gas is associated gas from oil fields, and a substantial share is reinjected to maintain reservoir pressure, approximately 65% in recent years according to national reporting (ANPG, 2023). This reinjection process, along with the use of gas turbines for power generation on deep-water facilities, can contribute significantly to CO<sub>2</sub> emissions.

. The FPSOs are connected to an underwater pipeline system that carries the associated gas to the operational LNG (liquefied natural gas) plant on the coast, where the gas is processed for export. The older, shallow-water facilities are not connected to this pipeline system and the associated gas is not commercialized but mainly reinjected or flared. Processing of oil occurs offshore for both types of facilities. The oil is then loaded to tanker ships for transport and sale.

Greenhouse gas emissions from Angola's O&G sector are significant due to the nature of extraction and processing activities. These emissions primarily originate from fugitive emissions, which are unintentional leaks from equipment and infrastructure; venting, which involves the intentional release of gas often due to safety reasons or the lack of infrastructure to capture and utilize associated gas; and flaring, which is the burning of excess gas that cannot be processed or sold.

Studies on CH<sub>4</sub> emission measurements from offshore platforms are limited but critical for accurate assessments. Some measurements have been conducted, such as ship-based measurements in the US Gulf of Mexico (Yacovitch et al., 2020), South-East Asia (Nara et al., 2014), the Chinese Bohai Sea (Zang et al., 2020), and the North Sea (Hensen et al., 2019; Riddick et al., 2019).

In contrast to ship-based measurements, the mobility of aircraft allows for sampling of emission plumes both horizontally and vertically, providing more detailed information on marine boundary layer conditions. To date, airborne measurements around offshore facilities have been conducted for example in the Sureste Basin, Mexico (Zavala-Araiza et al., 2021), the Gulf of Mexico (Gorchov Negron et al., 2020; Ayasse et al., 2022; Gorchov Negron et al., 2023; Biener et al., 2024), Alaska and California (Gorchov Negron et al., 2024), the Norwegian Sea (Roiger et al., 2015; Foulds et al., 2022), and the North Sea (Cain et al., 2017; Lee et al., 2018; Pühl et al., 2024).

The METHANE-To-Go Africa (MTGA) campaign conducted the first airborne methane measurements in West Africa. The study provides an empirical understanding of the magnitude and location of CH<sub>4</sub> emissions from the O&G industry in Angola. This publication is structured as follows: In Section 2, we describe the data, including airborne observations, inventory data, operator reporting, and satellite data, as well as the mass balance method used for the processing of the airborne data. In Section 3, we compare the different emission estimates for individual facilities and the entire Angola offshore sector. Section 4 gives a Discussion and Summary.

#### 2 Data and methods







The METHANE-To-Go project aims to better understand and quantify methane (CH<sub>4</sub>) emissions from the O&G sector, with a focus on offshore exploration. Within the METHANE-To-Go series, which is financed by the International Methane Emissions Observatory (IMEO) of the United Nations Environment Programme (UNEP) and the German Aerospace Center (DLR), airborne studies covering Europe (Italy, Bosnia, Serbia), the coal mining in Poland and the Middle East O&G production were conducted. This study focuses on the exploration and production activities off the coast of Angola. The DLR Institute of Atmospheric Physics (IPA) conducted a measurement campaign during three weeks in September 2022 (Section 2.1). Using an aircraft-based mass balance approach (Section 2.2) regional and facility-scale emissions are estimated. Results are compared to bottom-up emission inventories (Section 2.3), operator reporting (Section 2.4), and satellite data (Section 2.5).

#### 2.1 Airborne observational data

The DLR Falcon research aircraft was instrumented with a comprehensive suite of in-situ measurement systems for the detection of methane and related trace gases and measurement flights were performed along the coastal regions of Gabon, the Congo, and Angola. The flight strategy was optimized for deriving regional estimates of different (sub-) regions. Quantification of facility-scale emissions was possible on most days due to very favorable weather conditions. The MTGA campaign with the DLR Falcon aircraft took place between 5 and 26 Sep 2022. The campaign base was in Libreville, Gabon, and 15 measurement flights with a total of 60 flight hours were conducted. For the 10 flights in Angola, we made refueling stops at Luanda airport. The Dassault Falcon 20E-5 (Registration: D-CMET) is a twin-engine jet with unique modifications. They include air inlets on the roof, four underwing hardpoints for particle measurements probes, in-situ instruments for trace gas measurements inside the cabin, and a nose boom for pressure measurement. When flying at low altitudes <300 m, the DLR Falcon has a ground speed of around 110 m s<sup>-1</sup>, an endurance of 4 hours and can, thus, cover around 1,600 km during a single instrumented flight. Its instrumentation allows for precise meteorological measurements (Fimpel, 1991). High cabin temperatures of up to 50°C during the low-altitude low-speed measurement flights sometimes required improvisational cooling of some components, but in general the instrumentation worked well under the extreme conditions. Table 1 contains a list of the instruments installed and the parameters measured. Methane was measured with two instruments to provide redundancy for the primary target species of this campaign. We used the Quantum Cascade Laser Spectrometer (QCLS) methane data

(Kostinek et al. (2019) for emission estimation and, if not available, the Picarro data (Dischl et al. (2024); Harlass et al. (2024) was used. A comparison of both instruments has shown good agreement within their measurement uncertainties. The additional trace gases provide further insights into the sources of CH<sub>4</sub> emissions, e.g. CO<sub>2</sub> helps to distinguish between flaring and fugitive/venting emissions.

Figure 1 shows the flight tracks of the 10 flights in Angola. Each region was covered by at least two, sometimes three flights.

These were designed as either: 1.) survey, 2.) regional mass balance, 3.) or individual facility mass balance flights. The flight duration was around 4 hours with 1.5 hours used for the transfer to Libreville and Luanda and 2.5 hours spent in the measurement area. During this time between 6 and 18 facilities could be probed depending on their proximity.

Table 1: Instrument overview for MTGA campaign on the DLR Falcon.

| Instrument         | Species/<br>Parameter                                                                                 | Measurement frequency | Measurement Technique Reference                     |                                                   |
|--------------------|-------------------------------------------------------------------------------------------------------|-----------------------|-----------------------------------------------------|---------------------------------------------------|
| Aerodyne<br>QCLS   | CH <sub>4</sub> , C <sub>2</sub> H <sub>6</sub> ,<br><sup>13</sup> CH <sub>4</sub> , H <sub>2</sub> O | 0.5 s                 | Laser absorption spectroscopy                       | Kostinek et al. (2019)                            |
| Picarro<br>G2401-m | CH <sub>4</sub> , CO <sub>2</sub> , H <sub>2</sub> O                                                  | 2 s                   | Cavity ring-down spectroscopy                       | Dischl et al. (2024);<br>Harlass et al. (2024)    |
| IT-CIMS            | SO <sub>2</sub>                                                                                       | 1 s                   | Ion-trap chemical ionization mass spectrometry      | Speidel et al. (2007)                             |
| Thermo<br>SO2      | SO <sub>2</sub>                                                                                       | 1 s                   | Pulsed fluorescence analyzer                        | Luke (1997)                                       |
| ECO<br>Physics TR  | NO + NO <sub>y</sub>                                                                                  | 1 s                   | Chemiluminescence Technique                         | Harlass et al. (2024)                             |
| Aerosol            | vol. and non-vol. particles                                                                           | 1s                    | Condensation Particle Counters and<br>Thermodenuder | Feldpausch et al. (2006);<br>Dischl et al. (2024) |
| MET<br>package     | 3D-wind,<br>temperature,<br>humidity                                                                  | 10-100 Hz             | 5-hole pressure probe, PT100, dew point and Lyman-α | Fimpel (1991);<br>Bramberger et al. (2017)        |

Figure 1: (a) Flight tracks of the 10 scientific flights during MTGA and facility locations in Angola. The thin white line shows the division between shallow- and deep-water facilities. (b) Map of CH<sub>4</sub> observations in the boundary layer from the yellow flight in (a). The flight track is color-coded with the observed CH<sub>4</sub> concentration. Latitude and longitude values are not specified to protect the anonymity of the operators. Red diamonds show the locations of facilities received from the operators and the arrows the measured wind direction. The facility in the west was sampled at 3.5 and 9 km distance and several altitudes using a racetrack pattern.

#### 2.2 Mass balance method





The emissions of facilities or groups of facilities are determined using the airborne mass balance method (Mays et al., 2009; Turnbull et al., 2011; Karion et al., 2013; Pitt et al., 2019; Fiehn et al., 2020; Pühl et al., 2024). In short, the balance between the in- and outflowing mass of an imaginary box around the target is the mass emitted inside the box. These emissions are transported downwind while turbulence spreads them horizontally and vertically until the plumes are well-mixed from the surface (in our case the ocean surface) up to the planetary boundary layer height (PBLH). The aircraft tracks constrain the sides of the box. The upwind side covers the inflow into the box and the downwind side is used to determine the outflow of the box. The flight patterns are designed to cover the box, and especially the downwind side at different altitudes, with the highest track right above the PBLH. The flight tracks should ideally be perpendicular to the wind direction with a distance of 5 to 10 km to the source region allowing for reasonably well-mixed plumes and at the same time well-measurable enhancements (see below). There should also be sufficiently strong winds (>3 m s<sup>-1</sup>) to rule out accumulations of the observed species in the box. For the calculation we make the following assumptions: First, the wind speed, wind direction, emissions, background concentrations and PBLH remain constant over the sampling time. Second, there is no entrainment/detrainment into the free troposphere. Third, the lifetime of the species is much longer than transport and sampling times, which is true for CH<sub>4</sub> (lifetime ~9 years) and other long-lived greenhouse gases. Finally, the trace gas plume is well-mixed between the lowest

flight track and the ground. These criteria are most likely met in the early afternoon, when the PBLH has typically reached its maximum and atmospheric conditions are generally favorable for vertical mixing. However, local conditions may still limit mixing between the lowest flight altitude and the surface, particularly over water. The emissions F of all sources within the box are defined as the difference between inflow  $m_{in}$  and outflow mass fluxes  $m_{out}$ :

$$F = m_{out} - m_{in} = \sum_{i} (c_{out} - c_{in}) v_{i} A_{i}.$$
 (1)

Here  $m_{out}$  of transect i is determined from the measured enhancement above background concentrations in the downwind transect  $c_{out}$ , the wind speed perpendicular to the transect  $v_i$ , and the area  $A_i$  of the downwind side of the box.  $m_{in}$  is the mass of gas going into the box calculated from the background concentration  $c_{in}$  in the upwind transect. Since we did not always measure the upwind concentrations, background concentrations are determined from the 20 s of measurements taken immediately before and after the plume enhancement. The specific time interval is manually selected based on visual inspection of the data. If a second plume is close to the target plume, only the interval before or after the target plume is used for background determination. Summing up the fluxes from all horizontal transects i gives the mass flux through the entire plane. The concentration and wind observations between the flight transects at different altitudes are interpolated over the entire area of the outflow side. For interpolation we use the "layer"-method, which was also employed by Foulds et al. (2022), where the observed concentrations for each transect is assumed for the entire layer up to the middle between the next observation altitude. The lowest transect is extrapolated to the ground and the highest transect up to the PBLH assuming constant fluxes. We found no cases where the transect was still in the PBL, but no plume was detected any more, if it had been detected at lower altitudes. The uncertainty calculation and determination of the level of detection (LOD) is described in detail in Appendix A.

Most flights were designed to deliver a regional emission estimate using the mass balance method. The Angolan facilities were grouped into regions according to geographical proximity for the flight execution. Each region was covered at least twice with survey or regional mass balance flights. At least one dedicated mass balance flight was done for each region. The coastal regions were covered three times. Each measurement flight included vertical profiles to determine the PBLH before and after the downwind observations.

In general, the weather conditions were very favorable for mass balance calculations. This includes a clear boundary layer top and well-mixed plumes. The wind speed was stable throughout the measurements with a mean standard deviation of 0.36 m s<sup>-1</sup>. The primary wind was southerly and around 5 m s<sup>-1</sup>. This is visible in Figure 2a, which shows a map of one regional mass balance flight. The box pattern was slightly altered during the flight due to the plume being encountered further to the west than expected. All offshore facilities in the region were included in the mass balance box. The altitudes for the downwind transects were 220, 150, and 100 m, while the PBLH was at 350 m. Figure A2 shows example profiles of the variables through which the PBLH was estimated.

Some flights were designed as survey flights in order to scout all facilities of a region by flying a single transect downwind of the facilities, typically at an altitude in the middle of the PBL. Some parts of these flights also targeted a single facility with

transects at different altitudes and distances using a racetrack pattern to get a thorough emission estimate. A combination of a survey flight with a racetrack pattern can be seen in Figure 1b. With both flight patterns we can determine the emissions of individual facilities. Estimating emissions from survey flights has a higher uncertainty than the racetrack, because the vertical extent of the plume is less certain. A case study for the transects of the racetrack in Figure 1b is shown in Appendix A.

Often the emissions of individual facilities or groups of facilities could be extracted from the regional mass balance flights. This was possible if a clear background was observed between two plumes and the wind was steady enough in the region to deduce the potential source installation for each plume. Facilities were grouped according to the plumes that could be separated from the measurements. An example of this is given in Figure 2 with the plumes of group 1 and group 2.

210

215

200

Figure 2: (a) Box pattern flown around two groups of facilities at different altitudes with adjustments to capture the entire plume along the northern wall. Red diamonds show the locations of facilities and the arrows the wind direction. Red ovals show the groups of facilities for each observed plume. (b) Three transects of the northern wall at different altitudes with plumes 1 and 2 marked. Latitude and longitude values are not specified to protect the anonymity of the operators.

#### 2.3 Bottom up-emission inventory data

Methane and carbon dioxide emission data are available from global gridded bottom-up emission inventories like the Emissions Database for Global Atmospheric Research (EDGAR) from the European Commission JRC (2024), the Global Fossil Emissions Inventory v2 (GFEI) from Scarpelli et al. (2022) and the Copernicus Atmospheric Modeling Services Global anthropogenic emissions (CAMS-GLOB-ANT) (Granier et al., 2019; Soulie et al., 2023). In the EDGAR and CAMS inventories, the emissions are calculated using activity data, for O&G typically the amount of oil or gas produced, and an emission factor to derive regionally distributed emissions (European Comission JRC, 2024). The emissions are subject to

uncertainties because the regional and temporal variation of emissions is often not accounted for in the calculation. The GFEI uses the countries' total emissions reported to UNFCCC and distributes them geographically according to activity data or other proxies like population density. Angola's last report of emissions to UNFCCC was in 2005 with a total CH<sub>4</sub> emission of 960 kt y<sup>-1</sup> of which 487 kt y<sup>-1</sup> are supposed to originate from the energy industry with the rest attributed to agriculture and the waste sector (UNFCCC, 2023). Scarpelli et al. (2020) scaled these emissions to 2019, the year of their inventory emissions. We used EDGAR v8.0, CAMS-GLOB-ANT v6.1, both for the year 2022, and GFEI v2 data for the year 2019. All three inventorial emissions are available on a global 0.1° x 0.1° grid.

The geographical distribution of the inventorial CH<sub>4</sub> emissions, along with the locations of facilities provided by the operators in Angola (black dots), is displayed in Figure 3. The red lines denote the border towards Congo and Democratic Republic of Congo. The geographical distribution in EDGAR matches well with the locations of the facilities. Only very few facilities are not covered by emissions and few emitting regions do not include a facility. In the GFEI distribution, however, there are many pixels containing emissions that do not contain a facility. The emissions are more evenly distributed and do not always show hotspot pixels at facility locations. The CAMS-GLOB-ANT emission dataset shows only very few and small emission spots offshore, which are collocated with facilities. The CO<sub>2</sub> emission maps are shown in the Appendix Figure B1.

Figure 3: Maps of Angolan offshore region with CH4 emission from the three gridded emission inventories. The black dots denote the Angolan facilities and their size is relative to their oil production in 2021 as reported by the operators. The red lines show the borders to Republic of Congo and the Democratic Republic of the Congo. The red circles in the right figure show MARS methane emission detection locations (see Section 2.5). Latitude and longitude values are not specified to protect the anonymity of the operators.

## 2.4 Operator reported data







Currently, there are seven companies operating offshore oil facilities in Angola. In the following they are treated anonymously and are called Operator A to G. The Angolan offshore sector is organized in so-called blocks. This is the administrative unit used by the Angolan Agency of Petroleum and Gas (ANPG, Agência Nacional de Petróleo, Gás e Biocombustíveis). ANPG is the administrative agency responsible for reporting on the O&G sector to the Angolan Ministry of Mineral Resources, Oil and Gas (MIREMPET, Ministério dos Recursos Minerais, Petróleo e Gás Angola), which actively supported the present study and facilitated the communication with the operators. Both from ANPG and the operators we received detailed data on the O&G exploitation and environmental impacts in Angola.

The operators were informed about the upcoming measurement campaign to ensure safe flight operations, particularly regarding helicopter traffic and situational awareness. However, the specific timing of the measurements was not disclosed to the operators to prevent any potential operational changes that could bias the results.

Before the start of the measurement campaign we received the following facility information from the operators for 2021: name of facility, type of facility, block, year of commissioning, location, corresponding oil field, amount of oil produced, amount of gas produced, flare height, CH<sub>4</sub> emission, CO<sub>2</sub> emission. Following the campaign, we also received daily operational status and daily or monthly sums of: Oil produced, gas produced, gas burned, gas reinjected, fuel gas, lift gas, and gas exported to Angola LNG project (ALNG), emissions of CO<sub>2</sub> from fuel and flaring gas, CH<sub>4</sub> from flaring and fugitive for 2022. Where 2022 production or emission data were not provided, the annual data from 2021 or conversion ratios from other operators have been used to estimate emissions based on the amount of oil and gas produced.

The CH<sub>4</sub> and CO<sub>2</sub> emissions that we compare with our mass balance estimate were reported directly or calculated from reported proxies to ensure the best temporal resolution possible for each operator. Operator B and G directly reported all requested parameters in daily resolution. Operator E reported daily data of O&G production and fuel and flaring gas amounts and monthly emission data. Then daily emission data was calculated from the fuel and flaring gas amounts using the emission ratios per amount of fuel or flaring gas from operator B. The ratios used were 67.5 t CO<sub>2</sub> /mmscf (million standard cubic feet) for fuel gas and 74.5 t CO<sub>2</sub> /mmscf for flaring gas and 0.41 t CH<sub>4</sub> /mmscf from flaring. The calculated daily emission data fits with the reported monthly emissions within 1% for CO<sub>2</sub> and 20% for CH<sub>4</sub>. Operator D reported monthly fuel and flaring gas amounts. This was transferred into emission data using the same ratios from operator B and downscaled to daily data assuming temporally constant emissions. Operators A, C, and F did not report facility-level emissions for 2022 but did provide annual emissions data for 2021. We downscaled these 2021 facility values under the assumption of temporally constant emissions, which aligns well with the operator totals reported for September 2022.

## 2.5 Satellite data

The IMEO Methane Alert and Response system (MARS) draws data from nearly a dozen satellites to identify very large methane plumes and methane hotspots. IMEO scientists analyze the plumes and conduct further analysis using higher-resolution satellites(UNEP, 2024). In total, the data portal includes seven detected methane plumes from five facilities in the offshore region of Angola between November 2022 and August 2024. The locations are shown in Figure 3. All detected plume locations except for one detection collocate with Angolan offshore installation groups. The operator in the region of the unallocated detection has indicated that there are development projects in this region and the emission could result from a drilling ship or exploratory facility. Facility group F1\* has three detections including the strongest emission plume of  $9.19 \pm 4.60 \text{ t h}^{-1}$ .






275

#### 3 Results and discussion

## 3.1 Facility-scale emissions

During METHANE-To-Go Africa, we were able to determine fluxes for all Angolan offshore facilities, divided into 30 individual facilities and 10 facility groups. The 10 groups of facilities include 27 facilities taking the total count of investigated facilities to 57. There are two main types of offshore oil and gas facilities in Angola: older shallow-water platforms and newer deep- and ultra-deep-water installations. The shallow-water facilities are typically fixed platforms standing on the seabed. These often form multi-platform complexes, with additional satellite platforms functioning as wellheads. In such cases, the entire complex is considered a single facility. The largest of these includes up to 28 interconnected platforms or wellheads. There are 36 of these older, shallow-water facilities in Angola. In contrast, deep-water operations are conducted from Floating Production, Storage, and Offloading (FPSO) vessels, often converted oil tankers that are moored to the seafloor and connected via flexible pipelines to subsea wellheads. These FPSOs serve as both production and storage units, enabling oil extraction in areas far from shore. Our study includes 21 deep-water facilities.

In Figure 4 we display the methane emissions observed for the individual facilities and groups of facilities. The figure includes the number of facilities within each group in parentheses. Measurements have been repeated on different days and each observation is depicted separately. For 9 cases, the methane flux is set to zero, because it is below our theoretically lowest detectable flux. The lowest detectable flux is calculated for each mass balance individually and is between 0.8 and 10.3 kg h<sup>-1</sup> CH<sub>4</sub>. The total uncertainty includes the statistical uncertainty from trace gas and wind observations, the uncertainty of the background, and the uncertainty of the plume mixing height. For the latter, the number of transects flown through the plume is crucial. The mean total 1-sigma uncertainty for all mass balances is 36% mainly resulting from the uncertainty of the plume height, which contributes 76% to the total uncertainty. The statistical uncertainty contributes 4% and the background uncertainty 20%. For smaller fluxes, statistical uncertainty is the primary contributor, whereas for larger fluxes, the total

uncertainty is predominantly driven by uncertainties of the plume mixing height. The uncertainty assessment and lowest detectable flux calculation are described in detail in Appendix A.

For most facilities, the observed CH<sub>4</sub> emissions are similar on different days and show little temporal variability (Figure 4).

As an exception, on one day we captured a high-emission event of 10.4 t h<sup>-1</sup> from platform D3. On another day, though, the emissions from this facility were only 0.02 t h<sup>-1</sup>. The operator commented that nothing special happened on this platform during the time of the campaign. The plume is clearly attributable to D3 and was measured several times up to a distance of 75 km from the facility. The relevance of such an event heavily depends on its duration. Since we do not know about the duration in our case, we weighed both events equally, which results in mean emissions of the facility with a high total uncertainty of 100% (5.2 ± 5.2 t h<sup>-1</sup>). This approach integrates high-emission events into the average emissions of the entire Angolan offshore O&G sector. By capturing a substantial ensemble of measurements (87 mass balances across 57 facilities over 2.5 weeks) the intermittent nature of these high-emission events is accounted for in the overall assessment.

Another facility group with high methane emissions is from operator F with three measurements between 3.5 and 4.1 t h<sup>-1</sup> on the first two measurement days (four days apart) and one of 1.3 t h<sup>-1</sup> during the last measurement two days afterwards (F5\*). This shows that the high emissions were not consistent, but occurred for at least four days. Operator F also reported normal operations for all their facilities during the measurement flights. Both high-emission event facilities (D3 and F5\*) are shallow water platforms built in the 1980s (see Section 3.3). Unfortunately, we do not have production or emission data with daily resolution from operators D and F. This might have provided more insight into the causes of the variability in emissions, and the frequency or duration of these events.





The status of operations on each facility during the measurements was inquired from the operators directly. We received information from all operators except for operator C. Measurements during special operations are marked by a black outline in Figure 4. Special operations encompass anything that falls outside of standard procedures, e.g. shutdown, maintenance, offloading of oil to a tanker, gas export offline, and seawater system bond strand piping repair (injection offline). There were 7 cases of special operations reported for the total of 87 measurements. They do not coincide with high emissions, but rather show low emissions and one medium emission strength of 0.3 t h<sup>-1</sup> at facility B3 during offloading. It should be noted, however, that we visited only briefly for each measurement. The reported special operations may have taken place during a different time of day, causing the measurement to miss associated emission changes. Notably, during the high-emission events the operators reported normal operations and no venting, suggesting they were likely unaware of the emissions. Such emission are particularly difficult to mitigate, as a lack of awareness prevents timely detection and response.

When multiple measurements are available to calculate the average emission for a facility or group, we compute the mean emission  $\bar{F}$  and its uncertainty  $u_{\bar{F}}$  by combining the uncertainties of individual measurements  $u_{F_i}$  with the standard deviation  $\sigma$  of the individually observed fluxes. This ensures that both measurement uncertainty and day-to-day variability are captured in the reported uncertainties.

In general, higher emissions often originate from groups of older platforms in the shallow water regions (operators C, D, and F). One of these groups includes 28 facilities with all kinds of platforms like living, production, flaring and well jackets.

Deep and ultra-deep water facilities (operators B, E, G) tend to emit less with emissions ranging between the detection limit and 0.3 t h<sup>-1</sup>. This is further examined in Section 3.3. The onshore LNG terminal, which processes the natural gas from the deep-water facilities, exhibited low CH<sub>4</sub> emissions during both measurements.




Figure 4: CH<sub>4</sub> emissions determined from mass balance flights during MTGA for 30 individual facilities and 10 groups of facilities in Angola. The groups include the number of facilities within the group in parentheses. Markers at  $10^{-3}$  t h<sup>-1</sup> are below the theoretically lowest detectable flux and counted as zero. Emission estimates, where operators reported special operations, are marked by a black outline.

#### 3.2 Parameters impacting CH<sub>4</sub> emissions

Based on observations of other trace gases and data provided by the operators, we aim to identify the causes of the observed CH<sub>4</sub> emissions. We calculated the CO<sub>2</sub> fluxes for all facilities and facility groups in the same manner as the CH<sub>4</sub> fluxes. The individual results are shown in Appendix Figure A3. The observations show elevated CO<sub>2</sub> emissions mostly from operator C, who operates in the shallow-water region. The distinction between shallow- and deep-water facilities is not as clear as for CH<sub>4</sub> emissions, since there are also newer deep-water facilities from operators A, B, and G with emissions above 20 t t Figure 5a compares the CH<sub>4</sub> fluxes with the CO<sub>2</sub> fluxes for all facilities. High CO<sub>2</sub> emissions with low CH<sub>4</sub> levels suggest that the emissions likely result from combustion processes, such as flaring or stationary combustion in engines. Conversely, high CH<sub>4</sub> emissions with low CO<sub>2</sub> levels point to leaks or venting as the source. In Angola, the major CH<sub>4</sub> emission events seem to

originate from leaks or venting, as these plumes show minimal CO<sub>2</sub>. Facilities with high CO<sub>2</sub> emissions tend to release little CH<sub>4</sub>, indicating efficient flaring or turbine operations. During our campaign, we conducted targeted samples of seven flaring plumes in Angola, with detailed analyses of these flares to be published in a follow-up study. Figure 5b highlights CH<sub>4</sub> emissions in relation to the commissioning year of each facility, showing that the highest-emitting facilities tend to be those commissioned before 2000. These older facilities, located in shallow waters closer to the coast, are operated by companies C, D, and F. Figures 5c and 5d compare the measured CH<sub>4</sub> emissions to O&G production data from September 2022 or, if unavailable, the average production in 2021. High-production facilities, operated by A, B, E, and G, generally exhibit low emissions, while high-emitting facilities, mostly run by C, D, and F, show lower production volumes. This is particularly evident in gas production, due to the division of Angolan offshore facilities into those connected to the LNG plant and those that are not. The older facilities (C, D, and F) are not connected to the gas pipeline feeding the LNG plant. Instead, associated gas is reinjected, captured, or flared. Since this gas lacks economic value, potential leaks may not be closely monitored.


Figure 5: Observed CH<sub>4</sub> emissions of the Angolan facilities in relation to (a) CO<sub>2</sub> emissions, (b) year of commissioning of the facility (average for groups), (c) oil production in 2021/2022 in kbd<sup>-1</sup> (kilo barrel per day), and (d) gas production in 2021/2022 in mmscfd<sup>-1</sup> (million standard cubic feet per day).

Figure 6 shows the relations of CH<sub>4</sub> emissions to other trace gases measured on the Falcon aircraft during MTGA (CO, C<sub>2</sub>H<sub>6</sub>, SO<sub>2</sub>, and NO<sub>y</sub>). Carbon monoxide (CO) is an indicator for incomplete combustion processes and, thus, maybe malfunctioning flares or turbines. Some of the older facilities from operator C emit more than 200 kg h<sup>-1</sup> of CO along with their high CO<sub>2</sub> emission (see Figure 5a). The high-emission event on facility D3 is also accompanied by elevated CO emissions of 160 kg h<sup>-1</sup>. High CH<sub>4</sub> emissions from operator F are not accompanied by emissions of CO. Figure 6b depicts the strong correlation between CH<sub>4</sub> and C<sub>2</sub>H<sub>6</sub> emissions. This results from the common source of CH<sub>4</sub> and C<sub>2</sub>H<sub>6</sub> as components of the associated natural gas. The average molar C2/C1 emission ratio is  $15 \pm 6$  %. It is within the composition range of associated gas of 10-25 % (Xiao et al., 2008). Operator reports on the molar C2/C1 ratio of several facilities range from 5% to 20%. This matches with our observations. Figure 6c shows elevated SO<sub>2</sub> emissions from two onshore facilities of 37 kg h<sup>-1</sup> (A1) and 49 kg h<sup>-1</sup> (D6) without accompanying CH<sub>4</sub> or CO<sub>2</sub> emissions. The high-emission event from facility D3 is also accompanied by 20 kg h<sup>-1</sup> of SO<sub>2</sub>. This facility's gas is reported to contain H<sub>2</sub>S, which after combustion turns into SO<sub>2</sub>. Figure 6d displays the NO<sub>2</sub> emissions observed from the aircraft, with NO<sub>2</sub> serving as an additional tracer for combustion processes. Notably, NO<sub>2</sub> emissions show no correlation with CH<sub>4</sub> emissions, and for facility F5, varying levels of NO<sub>2</sub> emissions were detected even when CH<sub>4</sub> emissions remained at 4 t h<sup>-1</sup> across different days. This lack of correlation suggests that the CH<sub>4</sub> emissions are not from flaring but are likely due to venting or leakage.






due to reduced leakage and venting.

This analysis of our emission estimates, informed by operator data and observations of additional trace gases, allows for a more nuanced interpretation of our results. We find no correlation between CH<sub>4</sub> and CO<sub>2</sub> emissions, suggesting that they originate from different processes or sources on these facilities, namely fugitive emissions and venting for CH<sub>4</sub>, and combustion-related processes like flaring or power generation for CO<sub>2</sub>, CH<sub>4</sub> emissions are predominantly associated with older. low-production, shallow-water facilities, whereas high-production facilities, primarily newer FPSOs operating in deep and ultra-deep waters, exhibit relatively low CH<sub>4</sub> emissions despite their high output levels. By contrast, CO<sub>2</sub> emissions are elevated at both shallow-water platforms (notably operator C) and some deep-water facilities, as well as the LNG terminal. This distinction between shallow- and deep-water platforms can be linked to differences in facility type, age, and operational practices. Shallow-water facilities, generally built before 2000, are fixed structures mounted on the seabed and often form multi-platform complexes that include satellite platforms acting as wellheads. In such setups, the complex is treated as a single facility. The largest of these includes up to 28 interconnected platforms or wellheads. These older platforms typically rely on legacy infrastructure and may lack modern emission control technologies, leading to higher methane emissions from leaks or incomplete flaring. In contrast, deep-water operations are conducted from Floating Production, Storage, and Offloading (FPSO) vessels, converted oil tankers anchored to the seabed and connected to subsea wellheads via flexible pipelines. These FPSOs integrate production, processing, and storage, and generally feature more advanced systems for controlling emissions. In Angola, most FPSOs are also connected to an underwater pipeline system that carries the extracted gas to the LNG terminal on the coast. As such, their emissions tend to be dominated by CO<sub>2</sub> from combustion, with low levels of methane emissions

These findings suggest that methane emissions are not directly linked to production volume but rather to facility characteristics such as age, type, and maintenance status. Therefore, for bottom-up emission estimates, we propose that production data alone is not a reliable proxy for CH<sub>4</sub> emissions. Instead, facility age, type (shallow vs. deep water), and condition may serve as more informative indicators of methane emission strengths.

## 3.3 Comparison of observed emissions with operator reporting

The aircraft-based observations of average methane emissions from individual facilities or facility groups in Angola are shown in comparison with operator-reported emissions in Figure 7. Here we compare average observed emissions for each facility, calculated from the individual observations displayed in Figure 4. A broader comparison of total Angolan emissions with gridded inventory data and operator reporting is available in Section 3.6. We avoid comparing observations with gridded inventories at the facility level due to the limited number of samples per grid box, the significant uncertainties in attribution within global inventories, and the differing timescales between the two methods.

The operator data is described in detail in Section 2.4. For operators B, E, and G, the values are based on daily reports in September 2022. Operator D reported monthly fuel and flaring gas amounts for 2022, which were converted into emissions data. Operators A, C, and F did not report facility-level emissions for 2022 but provided annual data for 2021, which we downscaled. Only one operator directly reported fugitive emissions of CH<sub>4</sub>. All other operators only report CH<sub>4</sub> emissions derived from flaring. Reported CO<sub>2</sub> emissions result from flaring and fuel gas combustion.





The CH<sub>4</sub> emissions reported by operators (Figure 7a) are generally lower than the observed emissions, with maximum reported emissions at 1.5 t h<sup>-1</sup> for facility C3, where no significant emissions were measured. In contrast, for top-emitting facility D3 only 0.1 t h<sup>-1</sup> are reported by the operator as flaring emissions while fugitive emissions are not accounted for. Generally, emissions from older facilities (operators C, D, and F), from which we captured high-emission events, tend to be underestimated by operators, while emissions from newer, high-production facilities (operators A, B, E, and G) are often overestimated.

Figure 7: Observed average CH<sub>4</sub> and CO<sub>2</sub> emissions of the Angolan offshore facilities or groups of facilities compared to operator data consisting of daily or monthly reports for September 2022, or if not available downscaled annual data for 2021.

A comparison of observed CO<sub>2</sub> emissions with operator reports (Figure 7b) shows generally higher reported emissions than observed. CO<sub>2</sub> emissions result from the combustion of natural gas in gas turbines or flares, with gas use closely monitored and reported. However, CO<sub>2</sub> emissions reported by newer facilities (operators A, B, and G) are up to ten times higher than observed. This discrepancy could stem from the different timescales of our sampling and operator reports, especially since intermittent flaring often occurs during special operations. Notably, we did not observe any high CO<sub>2</sub> emission events from newer facilities. The observed atmospheric CO<sub>2</sub> enhancements for fluxes below 40 t h<sup>-1</sup> were often around the instrument uncertainty of 0.34 ppm. They were clearly distinguishable from the background fluctuations, though, and could be used for emission estimation.

Our measurements capture random snapshots of emissions, which are inherently variable across facilities, whereas operator reports reflect time-averaged emissions. These differences are expected at the facility level. However, they highlight two potential areas for improvements: collecting larger ensembles of observations to smooth short-term emission fluctuations, or enhancing the time resolution of operator reporting to enable more direct comparability at the facility level.

#### 3.4 Comparison with satellite data






The IMEO data platform provides a valuable resource for tracking methane emissions by listing and quantifying plumes observed via satellite (see Section 2.5). For the Angolan offshore region, seven methane plumes have been quantified from five distinct locations between November 2022 and February 2024 (Figure 8). All except one detection were allocated with groups of facilities in the shallow-water region of the Angolan offshore exploitation. The last detection is also in the shallow-water region, but nor connected to an existing platform. It might come from exploration activities. Emission estimates from groups C15\* and F5\* agree within uncertainties. Facility group F4\* showed lower emissions during aircraft observations than from the satellite detection. Group F1\* had three satellite detections. One of them has the highest satellite detected value with  $9.19 \pm 4.60 \text{ t h}^{-1}$ . Here, the satellite probably captured high-emission events from one of the facilities in group F1\*. This event is in the range of the high-emission event detected by aircraft from facility D3. The mean airborne emission observations of F1\*, however, were as low as  $0.43 \pm 0.22 \text{ t h}^{-1}$ .

While satellite observations are critical for identifying major emission sources, they cannot capture every emission event due to limited temporal resolution. Additionally, the current imaging satellites have detection limits around 1 t h<sup>-1</sup>. A comparison of airborne and satellite detections has also been discussed by Biener et al. (2024), who similarly concluded that space-based observations are effective at identifying methane super-emitter events. However, they note that differences in observed emission persistence are likely not due to changes in facility behavior between satellite and aircraft overpasses. Instead, these discrepancies are attributed to differences in detection thresholds and revisit times, with intermittent emissions potentially falling below satellite detection limits during some passes. This underscores the need for complementary ground-based or aircraft-based measurement campaigns to verify operator-reported emissions and detect fugitive emissions that may go unnoticed. Regular monitoring, particularly of high-risk facilities such as older or poorly maintained infrastructure, can help ensure accurate reporting and provide actionable data for mitigation.

Figure 8: Comparison of mean aircraft mass balance emission estimates with IMEO satellite detections. The second detection could not be attributed to a facility. Facility F1\* has three satellite detections.

#### 3.5 Comparison with other offshore production regions





We compared our CH<sub>4</sub> emission estimates from the Angolan offshore facilities with airborne measurements from other offshore regions in the world. Figure 9 presents a histogram of estimated fluxes from individual and grouped facilities in Angola, alongside the corresponding average emissions per facility. The average was calculated by dividing total observed emissions by the 57 facilities. Satellite structures were not counted separately. The mean CH<sub>4</sub> emission per Angolan facility is 0.30 t CH<sub>4</sub> h<sup>-1</sup>, but this value varies significantly depending on the facility's age. Facilities commissioned before 2000 (typically shallow-water platforms) exhibit substantially higher emissions, averaging 0.44 t CH<sub>4</sub> h<sup>-1</sup>, while newer facilities ( $\geq$ 2000, primarily deep-water FPSOs) emit an average of only 0.04 t h<sup>-1</sup>. This stark contrast reflects the influence of infrastructure type and age on methane emissions.

For comparison, the red lines in Figure 9 indicate the average emission rates per facility reported from offshore facilities in the Norwegian Sea, Southern North Sea, and the Northern Gulf of Mexico as synthesized by Pühl et al. (2024).

Emissions from nine Northern Gulf of Mexico facilities, including high-emitting central hubs comparable to Angola's legacy complexes, averaged 0.46 t CH<sub>4</sub> h<sup>-1</sup> (Gorchov Negron et al., 2020). In contrast, typical emissions from U.S. Gulf of Mexico platforms were reported as approximately 0.02 t h<sup>-1</sup> for shallow-water and 0.08 t h<sup>-1</sup> for deep-water platforms. These values indicate a pattern opposite to that observed in Angola, where deep-water platforms are the cleanest in terms of CH<sub>4</sub> emissions, whereas in the Gulf of Mexico, deep-water platforms contribute the highest emissions per facility. However, the dominance of older shallow-water facilities in platform counts means that they still drive total basin-wide emissions in both regions.

In Europe, reported average facility emissions are lower. In the Norwegian Sea, CH<sub>4</sub> emissions average 0.03 t CH<sub>4</sub> h<sup>-1</sup> per platform (Foulds et al., 2022), while in the Southern North Sea, emissions average 0.14 t h<sup>-1</sup> (Pühl et al., 2024). Facilities in these regions tend to be newer and predominantly produce gas, which likely contributes to their lower CH<sub>4</sub> emission rates. Further comparisons and discussions, including other studies and regions, are planned within the IMEO framework.

Figure 9: Histogram of our estimated mean CH4 fluxes of individual facilities and groups of facilities off the coast of Angola. The average value is marked in blue and calculated using the total number of facilities for each type. For comparison, average CH4 emission estimates for the Norwegian Sea, the Southern North Sea and the Gulf of Mexico (GOM) are indicated in red (Gorchov Negron et al., 2020; Pühl et al., 2024).

## 3.6 Total Angolan emissions and carbon intensities





The total Angolan offshore emissions of  $CH_4$  and  $CO_2$  derived by airborne mass balance during MTGA is calculated as the sum of the emissions from all facilities. The total error is the sum of all errors. This leads to total emissions of  $16.9 \pm 5.3$  t  $CH_4$  h<sup>-1</sup> and  $613 \pm 105$  t  $CO_2$  h<sup>-1</sup> including the high-emission events. The 10 t h<sup>-1</sup>  $CH_4$  emission event from platform D3 stands out, representing over half of Angola's total offshore emissions, highlighting the significance of such high-emission events. The frequency and duration of these events have a substantial impact on the country's overall emissions. The large ensemble of our measurements is deemed to capture the intermittent nature of high-emission events, incorporating them into the overall assessment.

Several gridded inventories also report offshore emissions for Angola (Figure 10). We summed the gridded inventories in the entire offshore region. The observed CH<sub>4</sub> emissions are only 20% and 22% of what is provided in the EDGAR v8.0 and GFEI v2 inventories, respectively. CAMS report very low emissions of only 1.4 t h<sup>-1</sup>. The International Energy Agency's (IEA)

Methane Tracker provides emissions data at the country level while distinguishing between various sectors. In Angola, total energy sector emissions are reported at 98 t h<sup>-1</sup>, with 86 t h<sup>-1</sup> attributed to the offshore sector. These emissions are divided into fugitives (19%), venting (74%), and flaring (7%). The IEA's estimates align with the ranges reported by EDGAR and GFEI. The overestimation of inventory CH4 emissions in Angola is indicative of a broader trend in which bottom-up inventories tend to overestimate emissions from offshore O&G production (Shen et al., 2023). In Angola, the overestimation of emission factors for newer Floating Production Storage and Offloading units (FPSOs) in inventories may contribute to discrepancies. These advanced facilities typically employ better technology and maintenance, resulting in emissions that are lower than those predicted by standard emission factors. Reassessing these factors could enhance inventory accuracy. Furthermore, operator-reported emission rates for 2021/2022 and the Second National Communication (SNC) of Angola to UNFCCC with the last emission report for 2015 (Republic of Angola, 2021) are nearly three times lower than observation-based estimates. Most operators focus on reporting methane emissions related to flaring and combustion, largely neglecting fugitive emissions. While we recognize the challenges in quantifying fugitive emissions without direct measurements, the gap between our observations and reported figures highlights the urgent need for regular monitoring at each facility to identify and mitigate fugitive emissions.

The observed  $CO_2$  emissions of  $613 \pm 105$  t  $h^{-1}$  are close to the EDGAR v8.0 inventory emission of 690 t  $h^{-1}$ . CAMS v6.1 emissions of 1412 t  $CO_2$   $h^{-1}$  and the operator reported emissions of 1389 t  $h^{-1}$  for 2022 are twice as high as the observed or EDGAR emissions. The offshore  $CO_2$  emissions originate from flaring or combustion. Operators can typically calculate their combustion emissions with precision, as the volume of burned gas is closely monitored and required to be reported. However, our flights may have missed a portion of  $CO_2$  emissions when they did not reach the upper half of the boundary layer, where hot flaring exhaust initially rises due to buoyancy, only dispersing at greater distances from the source. This effect, shown in Figure A1, may partly explain the discrepancy between measured and reported  $CO_2$  emissions, along with the challenges of aligning snapshot observations with emissions that vary over time due to operational shifts, maintenance, and other factors. Although we used a large set of measurements to capture a comprehensive picture, further sampling is necessary to better capture the temporal variability of the emissions. Comparing O&G production data received from the operators from September 2022 with 2021 data indicated no significant differences, suggesting that our measurement period reflects typical operations.

Figure 10: Total emissions of CH<sub>4</sub> and CO<sub>2</sub> for the Angolan offshore sector from observations, inventories, and reports. The International Energy Administration (IEA) methane tracker data was downloaded from their data portal (IEA, 2024). The Angola Second National Communication (SNC) to UNFCCC emissions are taken from the last emission report for 2015 (Republic of Angola, 2021).




Finally, we estimated the carbon and methane emission intensities of Angolan offshore oil and gas production using the total observed GHG emissions and reported production data for September 2022 (Figure 11). CH<sub>4</sub> emissions were converted to  $CO_2$ e equivalent using a global warming potential for 100 years (GWP<sub>100</sub>) of 29.8 (IPCC, 2021). The combined observed CH<sub>4</sub> and  $CO_2$  Angolan offshore emissions amount to 1116.6  $\pm$  270 t  $CO_2$ eq h<sup>-1</sup>. CH<sub>4</sub> and  $CO_2$  contribute roughly equally to this total.

According to operator reports, the average offshore production during September 2022 was 2118 kbd<sup>-1</sup> of oil and 2459 mmsfcd<sup>-1</sup> of gas. These gas volumes represent total produced gas, including amounts reinjected, flared, used for gas lift, and consumed as fuel. During the aircraft measurement days in September 2022, an average of 997 mmscfd<sup>-1</sup> of gas was exported to the LNG terminal, which is less than half of the total produced gas. An additional 1,428 mmscfd<sup>-1</sup> was reinjected, while the remainder was flared or used on-site. Since policymakers are primarily interested in emissions relative to the marketed oil and gas, we used only the volume of gas exported to the LNG terminal in our carbon intensity calculations.

Based on our airborne measurements, we estimate the overall carbon intensity of Angolan offshore oil and gas production to be  $3.4 \pm 0.8$  g CO<sub>2</sub>eq MJ<sup>-1</sup> for September 2022. A breakdown by facility type reveals notable differences. Older, shallow-water facilities (commissioned before 2000) exhibit a carbon intensity of 23.2 g CO<sub>2</sub>eq MJ<sup>-1</sup>, with methane contributing 66% of the total. In contrast, newer, deep-water facilities (commissioned in or after 2000) show a much lower intensity of 1.38 g CO<sub>2</sub>eq MJ<sup>-1</sup>, dominated by CO<sub>2</sub> emissions (92% of the total). This stark contrast highlights the improved emission performance of modern offshore operations, likely driven by better design, reduced fugitive methane emissions, and more efficient flaring or

combustion systems. These findings emphasize the potential for significant mitigation by upgrading or replacing aging infrastructure and targeting methane leaks in shallow-water platforms.

Our measurement-based carbon intensity estimate is considerably lower than existing inventory-based values. For instance, the EDGAR v8.0 dataset estimates Angola's carbon intensity at  $8.2 \pm 0.1$  g CO<sub>2</sub>eq MJ<sup>-1</sup> for the same time period. Similarly, Masnadi et al. (2018) estimate a carbon intensity of 7.5 g CO<sub>2</sub>eq MJ<sup>-1</sup> (range: 6.6–14.1) for Angola in 2015 using a bottom-up life cycle approach. These inventory estimates are more than twice as high as our measured average, highlighting potential overestimation in bottom-up methods, particularly for newer offshore infrastructure.





This finding aligns with recent results from the United States. Gorchov Negron et al. (2024) quantified carbon intensity across all U.S. offshore oil and gas production in 2021 and reported an average CI of 5.7 g CO<sub>2</sub>eq MJ<sup>-1</sup>. However, as in Angola, there is substantial variation by region and facility type. Deep-water platforms in the Gulf of Mexico (GOM) exhibited a low CI of 1.1 g CO<sub>2</sub>eq MJ<sup>-1</sup>, whereas older GOM federal shallow-water platforms reached 16 g CO<sub>2</sub>eq MJ<sup>-1</sup>, and GOM state shallow waters as high as 43 g CO<sub>2</sub>eq MJ<sup>-1</sup>. Offshore facilities on the North Slope of Alaska had a CI of 11 g CO<sub>2</sub>eq MJ<sup>-1</sup>.

These comparisons not only confirm the strong influence of infrastructure type and operational practices on emissions but also underscore the variable role of CH<sub>4</sub> and CO<sub>2</sub> across offshore regions. For example, the carbon intensity in Alaska's offshore production is dominated by CO<sub>2</sub> emissions, whereas GOM shallow-water platforms are primarily CH<sub>4</sub>-driven.

Figure 11: Carbon intensity of the Angolan offshore oil and gas sector including division with respect to CH<sub>4</sub> and CO<sub>2</sub> contributions and comparison with other studies (Masnadi et al., 2018; Liggio et al., 2019; Dixit et al., 2023; European Commission JRC, 2024; Gorchov Negron et al., 2024). Angolan facilities have been split according to their year of commissioning. Canadian oil sands (OS) results only give a range, while the total United States (US) offshore results from Gorchov Negron (GN) are also available for individual regions like the Gulf of Mexico (GOM) and the Alaskan North Slope.

The methane emission intensity divides total volumes of CH<sub>4</sub> emissions from both oil and natural gas value chains of operated assets by the total volumes of marketed natural gas. To convert the 997 mmscfd<sup>-1</sup> of gas exported to the LNG to a mass flux, we use the mean reported mole fraction of 78% of methane in the exported natural gas or the equivalent mean molar mass of 22 g/mol for Angolan natural gas. The observed methane intensity for September 2022 is then calculated to be 3.1%. Considering that Angola mainly produces oil with natural gas merely being a by-product we expect this high methane intensity. It is also caused by the high fraction of produced gas that is reinjected (53%) instead of exported. Shen et al. (2023) calculated the Angolan methane intensity to around 14% from methane emissions of 910 Gg a<sup>-1</sup>, which corresponds to 103 t h<sup>-1</sup>, and International Energy Agency gas production values for 2019.

#### 4 Summary





The dataset collected during the METHANE-To-Go Africa campaign is uniquely comprehensive, offering detailed measurements of CH<sub>4</sub> emissions from offshore O&G production along the West African coast, particularly Angola. Using an aircraft-based mass balance method, this analysis quantified emissions from all offshore facilities in Angola, with a focus on 30 individual facilities and 10 facility groups. Benefiting from stable wind conditions during flights, the mean 1-sigma uncertainty for methane emissions is 29%. Additional trace gas measurements, including CO<sub>2</sub>, CO, C<sub>2</sub>H<sub>6</sub>, SO<sub>2</sub>, NO<sub>y</sub>, and aerosol particles, provided further insights into sources of CH<sub>4</sub> emissions.

Our results show mainly consistent emission estimates across different days for most facilities, with minimal temporal variation. However, two facilities exhibited high-emission events of 10 and 4 t h<sup>-1</sup> on specific days, emphasizing the importance of capturing such events for total emissions estimates. Operator reports indicate that normal operations were in place during our observation periods, suggesting that these high emissions were unknown to them and likely due to leaks. Although satellite detections also reveal high-emission events from other facilities, the limited number of detections—just seven over three years—makes it difficult to assess the duration or frequency of these events. Enhanced operator awareness and more detailed reporting are essential to gain a clearer understanding of their impact.

Our findings suggest that significant CH<sub>4</sub> emissions in Angola stem from leakages or venting, as low CO<sub>2</sub> concentrations in plumes indicate limited flaring. Combustion processes in flares and turbines appear efficient, with high CO<sub>2</sub> emissions not being accompanied by elevated CH<sub>4</sub>. Our observations indicate that high CH<sub>4</sub> emissions primarily occur at older, low-producing facilities, while newer, high-producing FPSOs in deep and ultradeep water emit comparatively little methane. Production volume is again shown to be a poor estimator of emissions. To improve bottom-up emission estimates, we recommend considering facility age or maintenance status as factors that introduce additional risk for methane emissions, rather than purely relying on production volume as a proxy. Nevertheless, given the significant variability in asset design and

operation, measurements remain crucial. Regular measurements by e.g. operators should prioritize high-risk facilities, such as older ones.

Satellite detections of methane plumes offshore of Angola remain sparse, with seven observed plumes showing emission levels similar to our observations, including the enhanced emissions of the high-emission events that we observed at other facilities during the airborne campaign. The average observed airborne  $CH_4$  emission of Angolan offshore facilities was calculated at 0.30 t h<sup>-1</sup>, lower than that of the Gulf of Mexico (0.46 t h<sup>-1</sup>) but higher than the Norwegian Sea (0.03 t h<sup>-1</sup>) and Southern North Sea (0.14 t h<sup>-1</sup>).

Total emissions from Angolan offshore facilities were measured at  $16.9 \pm 5.3$  t CH<sub>4</sub> h<sup>-1</sup> during MTGA, representing 20% of EDGAR and 22% of GFEI inventory levels. Operator data for 2021 and 2022 underestimate CH<sub>4</sub> emissions by two-thirds relative to observations, with most operators only reporting flaring-related emissions and omitting fugitive emissions. We acknowledge the challenge in estimating fugitive emissions without measurements. However, this discrepancy underscores the importance of regular monitoring to detect and mitigate these emissions. Total observed CO<sub>2</sub> emissions are  $613 \pm 105$  t h<sup>-1</sup>, which is close to the EDGAR v8.0 inventory emission of 690 t h<sup>-1</sup>, but less than half of the CAMS v6.1 emissions of 1412 t CO<sub>2</sub> h<sup>-1</sup> and the operator reported emissions of 1389 t h<sup>-1</sup> for 2022. The significant differences between measured and reported emissions likely result from overestimated emission factors for newer FPSOs, underreporting of fugitive emissions, and the challenges of aligning snapshot observations with intermittent emission events.

The observed carbon intensity of Angolan offshore oil and gas stands at 3.4 ± 0.8 g CO<sub>2</sub>eq MJ<sup>-1</sup> for September 2022, less than half of previous estimates for the region. The observed methane intensity of Angolan gas is 3.1%, which is expectedly high for Angola's status as a country where gas is largely produced as a by-product of oil extraction. Importantly, a breakdown by platform age reveals a strong contrast in carbon intensities: older shallow-water facilities, typically commissioned before 2000, exhibit a significantly higher carbon intensity of 23.2 g CO<sub>2</sub>eq MJ<sup>-1</sup>, primarily driven by methane emissions (accounting for ~66%). In contrast, newer deep- and ultra-deep-water facilities, most of which are FPSOs commissioned after 2000, show a much lower carbon intensity of 1.38 g CO<sub>2</sub>eq MJ<sup>-1</sup>, with emissions dominated by CO<sub>2</sub> (~92%). This clear distinction underscores the impact of infrastructure age, design, and operational practices on emission profiles and highlights the potential climate benefits of modernization and stricter emissions control, especially in legacy infrastructure.

This study was conducted in close coordination with ANPG, MIREMPET, and local O&G operators. Results were presented to stakeholders in Luanda, Angola, in October 2022, where facility-specific feedback was provided. In response, operators expressed interest in continued monitoring, and ANPG is considering enhanced reporting requirements and emission reduction mandates for CH<sub>4</sub>.

By providing a uniquely detailed dataset on CH<sub>4</sub> and CO<sub>2</sub> emissions from Angola's offshore O&G industry, this study substantially enhances our understanding of emission sources and patterns in this oil and gas-producing region.

645

625

## Appendix A: Mass balance method uncertainties

665

670

The uncertainty of our emissions calculated with the mass balance method is combined from the statistical uncertainty, originating from the uncertainty in the measured parameters, and the systematic uncertainty, caused by the assumptions made for the method. The statistical uncertainty is calculated based on the measurement uncertainties, which are propagated through the mass balance equations using Gaussian error propagation. The uncertainties of the measured parameters are shown in Table A1. The uncertainty of the background concentration  $c_0$  is calculated as the standard deviation of the background interval. Here, we list the equations and respective uncertainty calculation of the parameters, which are marked with  $u_x$ :

Concentration enhancement: 
$$\Delta c_i = c_i - c_0 \implies u_{\Delta c_i} = \sqrt{u_{c_i}^2 + u_{c_0}^2}$$

Flux for each timestep 
$$i$$
:  $F_i = \frac{\Delta c_i \, v_i \, p_i \, dx_i \, M \, D_t}{R \, T_i} \quad \Rightarrow \quad u_{F_i} = F_i \, \sqrt{\left(\frac{u_{\Delta c_i}}{\Delta c_i}\right)^2 + \left(\frac{u_{v_i}}{v_i}\right)^2 \, + \, \left(\frac{u_{p_i}}{p_i}\right)^2 \, + \, \left(\frac{u_{D_t}}{dx_i}\right)^2 \, + \, \left(\frac{u_{D_t}}{D_t}\right)^2 \, + \, \left(\frac{u_$ 

Here M is the molar mass of the gas and R is the universal gas constant.

Flux for each transect t and statistical uncertainty: 
$$F_t = \sum_i F_i \implies u_{F_{stat}} = \sqrt{\sum_i u_{F_i}^2}$$

Table A1: Measurement parameters with instrument and measurement uncertainty during the MTGA campaign.

| Symbol                         | Parameter                                      | Instrument                     | Measurement uncertainty (1 $\sigma$ , 1s) |
|--------------------------------|------------------------------------------------|--------------------------------|-------------------------------------------|
| c <sub>i</sub> CH <sub>4</sub> | concentration of CH <sub>4</sub>               | Aerodyne QCLS                  | 1.55 ppb (Kostinek et al., 2019)          |
| $c_i$ CH <sub>4</sub>          | concentration of CH <sub>4</sub>               | Picarro G2401-m                | 1.25 ppb (after Klausner, 2020)           |
| $c_i \text{ CO}_2$             | concentration of CO <sub>2</sub>               | Picarro G2401-m                | 0.34 ppm (after Klausner, 2020)           |
| $c_i C_2 H_6$                  | concentration of C <sub>2</sub> H <sub>6</sub> | Aerodyne QCLS                  | 0.24 ppb (Kostinek et al., 2019)          |
| v                              | horizontal wind speed (u and v direction)      | Flow angle sensor              | 4 % (Giez et al., 2022)                   |
| p                              | pressure                                       | Flow angle sensor              | 50 Pa (Bramberger et al., 2017)           |
| x                              | horizontal distance                            | IGI GNSS/IMU:<br>Compact FOG-I | 0.02 m (AEROcontrol)                      |
| D                              | depth of each transect layer                   | IGI GNSS/IMU:<br>Compact FOG-I | 0.20 m (AEROcontrol)                      |
| T                              | temperature                                    | PT100                          | 0.5 K (Fimpel, 1991)                      |

The systematic uncertainty for our mass balance approach mainly consists of the uncertainty of the background concentrations and the plume mixing height uncertainty. We estimated the uncertainty of these two parameters for each transect individually. For the background value uncertainty, we calculated the flux using an average background concentration from an upwind flight track if an upwind flight track was available. The uncertainty was then defined as the difference between the flux using upwind background and the standard background from the edges of the plume. This was available for 16 of the total 99 mass balances. The average uncertainty due to the background was 10% of the total flux for these cases. Therefore, we added a background

uncertainty of 10% to all mass balances without upwind data. Figure A1 shows six examples of downwind concentrations for selected plumes. The CH<sub>4</sub> background concentration including uncertainty as shading is shown as red line and the upwind background (if available) as black horizontal line.

Figure A1: Downwind concentrations for six selected plumes. Background values are in the gray shaded areas. The CH<sub>4</sub> background concentration including uncertainty is shown as red line with shading and the upwind background value (if available) as black horizontal line.

The plume mixing height uncertainty relates to our assumption that every observed plume reaches from the ocean surface to the top of the PBL. We measured the plumes at distances between 4 and 15 km from the source, with most flight tracks between 5 and 10 km distance. Based on our experience, this distance should be a good compromise: Sampling further away from the source increases uncertainties related to measuring lower enhancements. This is accounted for by Gaussian error propagation. Sampling closer to the source introduces uncertainties related to incomplete mixing, which we take into account through the plume mixing height uncertainty. The cases where we have several transects at different altitudes showed average concentrations in the middle of the PBL, but we never measured directly at ocean surface nor at PBLH level (see Section 2.2). We assumed that the plume mixing depth toward the ocean surface is uncertain up to half the height of the lowest flight transect. Additionally, the plume mixing height is uncertain between middle of the highest transect and the PBLH up to the

PBLH. In this way we also accounted for the uncertainty in the determination of the PBLH. The method includes that the calculated flux becomes more uncertain the fewer transects we flew. The single transect mass balances were targeted to be flown in the middle of the boundary layer. An example of the uncertainties for single and multiple transect calculations is shown in Figure A1 for the racetrack flight pattern in Figure 1b. The transect at 250 m falls within the average CH<sub>4</sub> emission range, with the lower three transects below and the upper ones above the mean emissions. This behaviour was observed in several cases, but sometimes also with reversed profile. From this, we conclude that single transect mass balances are reliable when conducted in the middle of the planetary boundary layer (PBL). Below 150 meters, they should be used with caution, and above the middle level, emissions may be overestimated.

For each transect, the three individual uncertainties (statistical, background concentration, and plume height) are summed in quadrature to derive the emission uncertainty.

$$u_{F_t} = \sqrt{u_{F_{stat}}^2 + u_{F_{bg}}^2 + u_{F_{ph}}^2}$$

Finally, the total flux and total uncertainty are calculated as sum of all transect fluxes and uncertainties:

$$F = \sum_t F_t \implies u_F = \sum_t u_{F_t}$$

In the case of having several measurements for a facility or group of facilities, we also calculate the mean emission per facility  $\overline{F}$  from the single measurements. The uncertainty of the mean facility emissions  $u_{\overline{F}}$  then is a combination of the uncertainties of the single measurements  $u_{F_i}$  and the standard deviation  $\sigma$  of these measurements with n being the number of

single measurements: 
$$u_{\bar{F}} = \sqrt{\left(\frac{1}{n}\sum_{i}u_{F_{i}}\right)^{2} + \left(\frac{\sigma}{\sqrt{n}}\right)^{2}}$$


Figure A1: Mass balance estimates for each single transect including uncertainties as markers and the emission estimate using all transects and the layer method with uncertainty in red. The PBL was at 450 m. Transects in the middle of the PBL show good agreement with the overall emission estimate.

Generally, the PBLH was well defined and was detected from the maximum in the gradient or manually from potential temperature, water vapor mixing ratio and vertical wind during profile flights before or after the mass balance flight pattern. An example is shown in Figure A2.

Figure A2: PBLH determination from gradients in potential temperature  $\Theta$ , water vapor mixing ratio and vertical wind for one profile during flight 13b. Profiles of CH<sub>4</sub> and CO<sub>2</sub> are also displayed.




The theoretically lowest detectable flux was defined as the smallest signal detectable with 95% significance (2σ) over three consecutive time steps in each transect of the mass balance calculation. The final lowest detectable flux is then the sum of the lowest detectable fluxes across all transects. This threshold is primarily influenced by measurement and background uncertainties: if background variation exceeds measurement uncertainty, signal enhancements remain undetectable, precluding flux calculations. Wind speed and PBLH are also factored in, as they affect plume mixing within the boundary layer. The calculated lowest detectable fluxes are between 0.8 and 10.3 kg h<sup>-1</sup> CH<sub>4</sub> and between 406 and 7116 kg h<sup>-1</sup> CO<sub>2</sub>. The lowest values occurred for one measurement close to the coast, where the wind speed was only 1.3 m s<sup>-1</sup>, the background concentration uncertainty at 0.96 ppb CH<sub>4</sub> and 0.04 ppm CO<sub>2</sub>. The observed enhancement in this case was 10 ppb CH<sub>4</sub> and 0.40 ppm CO<sub>2</sub>, resulting in a flux of 27 kg h<sup>-1</sup> CH<sub>4</sub> and 2430 kg h<sup>-1</sup> CO<sub>2</sub> at 340 m PBLH. For CO<sub>2</sub>, this is close to the instrument uncertainty, but due to the low background concentration uncertainty, the plume is clearly distinguishable and counted. In other cases, the CO<sub>2</sub> background is more variable and plumes cannot be clearly distinguished. Of the total 85 mass balances, 9 were below the detectable flux for CH<sub>4</sub> and 13 for CO<sub>2</sub>.

## Appendix B: CO<sub>2</sub> gridded emission inventories and observed fluxes


Figure B1 shows the  $CO_2$  emission distribution of EDGAR v8.0 and CAMS v5.3 for the year 2022. This distribution matches well with the location of the offshore facilities.

Figure B1: Maps of annual  $CO_2$  emissions for the year 2022 from the two emission inventories EDGAR v8.0 and CAMS-GLOB-ANT v5.3.

Figure B2: CO<sub>2</sub> emissions determined from mass balance flights during MTGA for 30 individual facilities and 10 groups of facilities in Angola. The groups are marked with \*. Emission estimates, where operators reported special operations, are marked by a black outline.

**Author contributions:** AF prepared the campaign, collected, analyzed, and interpreted the data and wrote the manuscript. ME coordinated the measurements during the campaign phase and developed the mass balance routine. TB and MP measured and analyzed the CH<sub>4</sub>, CO<sub>2</sub>, CO, NO<sub>y</sub> and C<sub>2</sub>H<sub>6</sub> data. KG led the weather forecast, flight planning and execution operations. HA and LE measured and analyzed SO<sub>2</sub>. GN, FS, and DS measured and analyzed the aerosol data. RB provided tracer forecast simulations for the flight planning. GDAV, WNC, DLZ, MX, PC led communication with the operators, collected and provided their data and helped interpreting it. AR developed the research idea and led the MTGA project. All authors contributed to the interpretation of the results and the improvement of the manuscript.

**Competing interests:** The authors have no competing interests.



Acknowledgement: The authors thank DLR-FX for the campaign cooperation, especially the pilots Michael Grossrubatscher and Götz Hieber, in-flight technicians David Woudsma and Georg Reiser, the sensor group of Andreas Giez, Martin Zöger, Kevin Raynor, and Andreas Numberger, project manager Oliver Paxa, flight operations officer Frank Probst, technicians Stephan Storhas and Robert Uebelacker. We also thank Larissa Mengue and Mpiga Assele Ulrich from the Agence Gabonese d'Etudes aet d'Observations Spatiales (AGEOS) for their support of the campaign at our base in Gabon, the Gabonese Airforce for the opportunity to use their Libreville facilities for our flight operations. The authors acknowledge ECCAD for archiving and distributing the CAMS emission inventories.

Funding information: This project has been funded through UNEP's International Methane Emissions Observatory (IMEO).

Data accessibility: The measurement data is stored on the HALO database (https://halo-db.pa.op.dlr.de/) and publicly available at zenodo (Fiehn et al., 2025).

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
