# Peer review of "Airborne quantification of Angolan offshore oil and gas methane emissions"

_EGUsphere, 2025_

## Referee Comment (RC1)

**Review of EGUsphere Manuscript (#egusphere-2025-635)**

"Airborne Quantification of Angolan Offshore Oil and Gas Methane Emissions" by A. Fiehn et al.

**General Comments**

This study presents aircraft-based measurements of methane emissions from offshore oil and gas (O&G) operations along the West African coast, using the mass balance approach. The authors demonstrate that bottom-up inventories significantly overestimate emissions, whereas operator-reported values underestimate them. The findings highlight the need for regular airborne and in-situ measurements to improve methane emission quantification and support mitigation efforts. Given Angola's commitment to the Global Methane Pledge, these empirical assessments are critical for guiding policy and regulatory strategies aimed at reducing O&G sector emissions.

Overall, I think this study contributes valuable insights into offshore methane emissions, an underrepresented source in global emission inventories. It underscores the importance of empirical verification and identifies key areas for targeted mitigation strategies. The paper is well-written in general and may be accepted for publication after addressing the following specific comments.

**Special Comments**

1. **Abstract:** Could the lower observed methane emissions from the aircraft measurements compared to EDGAR and CAMS inventories be due to possible unmeasured methane sources in the region?
2. **Line 36:** Suggest removing "long-lived". The methane lifetime of approximately a decade is relatively short, especially compared to CO2, which persists for centuries. This is also conflicts with the phrase "short lifetime" in Line 39)
3. There is some repeated information, e.g., Line 34 and Line 38 about "the second most significant (important) long-lived anthropogenic greenhouse gas". Please make it concise and combine the first 3 lines into the paragraph starting in Line 36.
4. **Line 43:** The 22% figure is repeated from Line 35. Please consolidate for clarity.
5. **Figure 1:** The methane enhancements in downwind plumes are on the order of 5 ppb. What are the instrumental precision values, and how do they contribute to the overall uncertainty in the mass balance emissions estimate?
6. **Line 139:** How did the authors ensure that methane plumes from facilities are horizontally and vertically well-mixed from the surface?
7. **Line 144:** Is a 5–10 km distance from the source sufficient for plumes to be vertically well-mixed? This is a key concern, as later discussions suggest that plumes are likely not well mixed, which could introduce additional uncertainty.
8. **Lines 149–150:** The statement "These criteria are most likely to be met in the early afternoon, when the PBLH has reached its maximum" may not be entirely valid, particularly regarding the last criterion: "the trace gas plume is well-mixed between the lowest flight track and the ground."
9. **Methane Emission Variability:** while relatively consistent CH4 fluxes were observed for some facilities across different days, flight-to-flight variations — such as the two

high-emission events from two facilities — are not explicitly accounted for the uncertainty analysis.  This should be addressed.

10. **L.539-542:** this paragraph seems more appropriate for the Acknowledgement section.

11. **Appendix A:** the authors provided a thorough analysis of mass balance method uncertainties, covering statistical errors, background concentration, and plume height.  However, another two key uncertainties should be addressed or at least mentioned: how well the $CH_4$ plume is mixed within the planetary boundary layer (PBL) and day-to-day variability of emissions.  Fig. 1 A shows a factor of 5-6 variation in fluxes at different altitudes, suggesting that the plumes were not very well mixed at typical distances of 5-15 km downwind from the facilities.  This is back to my earlier comment that this distance range might not be enough for plumes to be well mixed within the PBL.  This is particularly relevant for surveys with fewer transects, as insufficient sampling could lead to larger uncertainty, i.e., less transects will have large uncertainty due to not being well mixed.

12. **L.620-625:** Fig. B1: the caption and text refer to CO2 emissions, but the figure scale label shows CH4 fluxes.  Please ensure consistency.

13. **Temporal Variability**: The study was conducted over a three-week period, which may not fully capture seasonal variations in emissions.

14. **Limited Facility Access**: While the study covered a significant number of facilities, more extensive coverage across different offshore production environments could provide a more comprehensive picture.

---

## Author Comment (AC1)

**Answer to referee comment 1 for "Airborne quantification of Angolan oil and gas methane emissions"**

*We would like to thank the reviewer for the suggestions to improve the manuscript. Below you find our answers to their comments. The reviewer's comments are written in normal font, our answers in italics.*

**General Comments**

This study presents aircraft-based measurements of methane emissions from offshore oil and gas (O&G) operations along the West African coast, using the mass balance approach. The authors demonstrate that bottom-up inventories significantly overestimate emissions, whereas operator-reported values underestimate them. The findings highlight the need for regular airborne and in-situ measurements to improve methane emission quantification and support mitigation efforts. Given Angola's commitment to the Global Methane Pledge, these empirical assessments are critical for guiding policy and regulatory strategies aimed at reducing O&G sector emissions.

Overall, I think this study contributes valuable insights into offshore methane emissions, an underrepresented source in global emission inventories. It underscores the importance of empirical verification and identifies key areas for targeted mitigation strategies. The paper is well-written in general and may be accepted for publication after addressing the following specific comments.

*We thank the reviewer for this positive assessment and will address the specific comments below.*

**Specific Comments**

1. **Abstract:** Could the lower observed methane emissions from the aircraft measurements compared to EDGAR and CAMS inventories be due to possible unmeasured methane sources in the region?

*There should not be any unmeasured sources in the region. We received a complete list of offshore assets from the operators and covered all of them during our flights. We used the gridded inventory data to calculate offshore emissions to avoid any sources onshore. Maybe in some cases the attribution of emissions between onshore and offshore in the inventories is not precise. We believe that to be the case for CAMS-GLOB-ANT emissions, which are much too small in the offshore region.*

2. **Line 36:** Suggest removing "long-lived". The methane lifetime of approximately a decade is relatively short, especially compared to $CO_2$, which persists for centuries. This is also conflicts with the phrase "short lifetime" in Line 39)

*We removed this.*

3. There is some repeated information, e.g., Line 34 and Line 38 about "the second most significant (important) long-lived anthropogenic greenhouse gas". Please make it concise and combine the first 3 lines into the paragraph starting in Line 36.

*We also removed repeated information here.*

4. **Line 43:** The 22% figure is repeated from Line 35. Please consolidate for clarity.

*We deleted the sentence in line 35.*

5. **Figure 1:** The methane enhancements in downwind plumes are on the order of 5 ppb. What are the instrumental precision values, and how do they contribute to the overall uncertainty in the mass balance emissions estimate?

*Measurement uncertainties for each parameter are given in Table A1 in the Appendix. These measurement uncertainties propagate through the mass balance calculation with Gaussian error propagation to form the statistical uncertainty, which ranges from 1.3 to 58.4 kg h$^{-1}$ and contributes 4.3% to the total uncertainty of the fluxes.*

6. **Line 139:** How did the authors ensure that methane plumes from facilities are horizontally and vertically well-mixed from the surface?
*There is no way to ensure this completely. We kept a minimum distance of 5 km to the facilities to allow for sufficient vertical mixing. Also, we tried to cover the plume with as many tracks as possible and the lowest track close to the ocean surface. We analyzed a couple of cases with several transects (see Appendix) and found that emissions calculated from transects in the middle of the PBL generally were within the emissions estimate using all transects. Thus, we also used single-transect cases for flux estimation accounting for the increased uncertainty through the "plume mixing height uncertainty".*

7. **Line 144:** Is a 5–10 km distance from the source sufficient for plumes to be vertically well-mixed? This is a key concern, as later discussions suggest that plumes are likely not well mixed, which could introduce additional uncertainty.
*There is a tradeoff: Sampling further away from the source increases uncertainties related to measuring lower enhancements. Sampling closer to the source introduces uncertainties related to incomplete mixing. The distance at which a plume is well-mixed depends on several factors including atmospheric stability, wind speed, and plume temperature. We acknowledge that not all measurements might have taken place in the best possible distance. We account for the possibility of not well mixed plumes via the "plume mixing height uncertainty". This has been clarified in Section 2.2 and in the appendix.*

8. **Lines 149–150:** The statement "These criteria are most likely to be met in the early afternoon, when the PBLH has reached its maximum" may not be entirely valid, particularly regarding the last criterion: "the trace gas plume is well-mixed between the lowest flight track and the ground."
*Thank you for pointing this out. We agree that the assumption of a well-mixed trace gas plume between the lowest flight track and the ground may not universally hold, even in the early afternoon. While it is true that the planetary boundary layer height (PBLH) typically peaks during this period, we acknowledge that local meteorological conditions (e.g., stratification, coastal effects, or shallow boundary layers over water) can limit the degree of vertical mixing, particularly over the ocean or complex terrain.*
*In our case, we have addressed this limitation by including it in the uncertainty assessment. We will revise the sentence in the manuscript for greater precision. For example, we could modify it to: "These criteria are most likely to be met in the early afternoon, when the PBLH has typically reached its maximum and atmospheric conditions are generally most favorable for vertical mixing. However, local conditions may still limit mixing between the lowest flight altitude and the surface, particularly over water."*

9. **Methane Emission Variability:** while relatively consistent CH4 fluxes were observed for some facilities across different days, flight-to-flight variations — such as the two high-emission events from two facilities — are not explicitly accounted for the uncertainty analysis. This should be addressed.
*We agree that variability between flights, including high-emission events, should be reflected in the uncertainty analysis. When multiple measurements are available for a facility or group, we compute the mean emission $\bar{F}$ and its uncertainty $u_{\bar{F}}$ by combining the uncertainties of*

*individual measurements $u_{F_i}$ with the standard deviation $\sigma$ of the observed fluxes. The uncertainty is calculated as:*

$$u_{\bar{F}} = \sqrt{\left(\frac{1}{n}\sum_i u_{F_i}\right)^2 + \left(\frac{\sigma}{\sqrt{n}}\right)^2}$$

*This ensures that both measurement uncertainty and day-to-day variability are captured in the reported uncertainties. The description is in the appendix and we added the information to the section.*

10. **L.539-542:** this paragraph seems more appropriate for the Acknowledgement section.
*We rephrased this paragraph to make it sound less like an acknowledgement: "This study was conducted in close coordination with ANPG, MIREMPET, and local oil and gas operators. Results were presented to stakeholders in Luanda, Angola, in October 2022, where facility-specific feedback was provided. In response, operators expressed interest in continued monitoring, and ANPG is considering enhanced reporting requirements and emission reduction mandates for $CH_4$."*

11. **Appendix A:** the authors provided a thorough analysis of mass balance method uncertainties, covering statistical errors, background concentration, and plume height. However, another two key uncertainties should be addressed or at least mentioned: how well the $CH_4$ plume is mixed within the planetary boundary layer (PBL) and day-to-day variability of emissions.
*The uncertainty associated with how well the $CH_4$ plume is mixed within the planetary boundary layer (PBL) is indeed addressed through what we previously referred to as "plume height uncertainty." To better reflect this, we have renamed it "plume mixing height uncertainty" to avoid ambiguity. Regarding day-to-day variability, this is incorporated into the uncertainty of the mean facility emissions by including the standard deviation of individual measurements alongside their respective uncertainties.*

Fig. A1 shows a factor of 5-6 variation in fluxes at different altitudes, suggesting that the plumes were not very well mixed at typical distances of 5- 15 km downwind from the facilities. This is back to my earlier comment that this distance range might not be enough for plumes to be well mixed within the PBL. This is particularly relevant for surveys with fewer transects, as insufficient sampling could lead to larger uncertainty, i.e., less transects will have large uncertainty due to not being well mixed.
*We agree that significant variation in fluxes across different altitudes, as shown in Fig. A1, suggests that plumes are not always fully mixed within the planetary boundary layer (PBL) at distances of 5–15 km downwind. This is particularly relevant for cases with only a few transects, where insufficient sampling may result in larger uncertainties.*
*Our uncertainty estimate—now renamed plume mixing height uncertainty to better reflect this issue—explicitly accounts for incomplete mixing. The number and vertical coverage of transects are key factors: fewer transects or strong vertical gradients lead to increased uncertainty in the flux estimate.*
*Figure A1 illustrates this with an example from a racetrack flight pattern (Fig. 1b), where fluxes from seven transects show a factor of 5–6 variation. The transect at 250 m aligns closely with the average $CH_4$ emission across all transects, with lower transects underestimating and higher ones overestimating the mean. This pattern, while not universal, was commonly observed. From this, we conclude that single-transect mass balances can be*

*considered reliable when conducted near the middle of the PBL. Measurements below 150 m should be treated with caution, while those above the midpoint may lead to overestimation. This altitude-dependent uncertainty is reflected in our reported total flux uncertainties and is one reason why we prioritize flight designs with multiple vertical passes where possible.*

12. **L.620-625:** Fig. B1: the caption and text refer to CO2 emissions, but the figure scale label shows CH4 fluxes. Please ensure consistency.
*We changed the label to CO2 emission.*

13. **Temporal Variability**: The study was conducted over a three-week period, which may not fully capture seasonal variations in emissions.
*We acknowledge that our three-week measurement period cannot fully capture potential seasonal variability in methane emissions. However, by conducting a large number of measurements during this time and comparing operator-reported oil and gas production data from September 2022 with that of 2021, which showed no significant differences (mentioned in Section 3.6), we conclude that our observations are representative of typical operational conditions. Nonetheless, we agree that further sampling over longer timeframes would be valuable to more comprehensively characterize temporal variability.*

14. **Limited Facility Access**: While the study covered a significant number of facilities, more extensive coverage across different offshore production environments could provide a more comprehensive picture.
*During the METHANE-To-Go Africa campaign, we determined methane fluxes from all offshore oil and gas facilities in Angola. This included 30 individual facilities and 10 facility groups. This full coverage is clarified  at the beginning of Section 3.1 and visually represented in Figure 1. While broader regional or global comparisons would benefit from extended coverage across diverse offshore production environments, our dataset provides a comprehensive snapshot of emissions from the entire Angolan offshore sector.*

---

## Author Comment (AC2)

**Answer to referee comment 2 for "Airborne quantification of Angolan oil and gas methane emissions"**

*We would like to thank the reviewer for the detailed comments and suggestions to improve the manuscript. Below you find our answers to their comments. The reviewer's comments are written in normal font, our answers in italics.*

**Overview**

The authors present an important addition to the literature on offshore oil and gas GHG emissions by expanding observations to Angola for the first time. They conducted an extensive airborne campaign and carefully calculated mass balance observations that they compare with inventories. The implications of their results can be better highlighted with a more careful presentation of their findings and a more integrative comparison with the offshore literature. This includes (A) a more thorough description of the shallow water facility clusters to determine if they are independent facilities or groups of dependent facilities as previously described in other papers, (B) propagating their shallow vs deep water trends into the final figures, including in their discussion on carbon intensity, and how these trends compare to the literature, and (c) raising the profile of $CO_2$ in their story as it appears to contribute to over half of the $CO_2$eq emissions, including in the abstract, body text, figures, and maybe a new figure.

*We thank the reviewer for this positive assessment and very good suggestions. We will explain our manuscript changes in the specific comments below.*

**Comment 1**

This study focuses on methane, but estimates and evaluates carbon dioxide fluxes as well. In addition, $CO_2$ is combined with $CH_4$ into an estimate of carbon intensity. This is relatively rare in the airborne oil and gas literature as this field has historically assumed that O&G production field methane emissions are more important (with GWP scaling) and less well known. This is not necessarily always the case based on the few studies that measure carbon dioxide (see below). In fact, this studies' $CO_2$ flux of 613 t/h and $CH_4$ flux (with GWP scaling 16.8 X 29.8 = 476.8 t $CO_2$eq/h) shows that $CO_2$ and $CH_4$ both contribute to the carbon intensity of the basin with $CO_2$ contributing a possibly larger fraction. However, this is not shown or explicitly stated, but is an interesting result.

I suggest the authors consider highlighting the $CO_2$ side of their story more in the introduction, show what fraction of the carbon intensity is driven by $CH_4$ and $CO_2$ (for both shallow and deep water), and perhaps, if they think it is within scope, compare with other carbon intensity estimates in other basins. Perhaps a final carbon intensity figure may help communicate this.

Relevant Literature on O&G $CO_2$ and carbon intensity

Liggio J. et al 2019, Measured Canadian oil sands $CO_2$ emissions are higher than estimates made using internationally recommended methods, Nature Commun. 10 1863: https://www.nature.com/articles/s41467-019-09714-9

Wren et al 2023, Aircraft and satellite observations reveal historical gap between top–down and bottom–up $CO_2$ emissions from Canadian oil sands https://academic.oup.com/pnasnexus/article/2/5/pgad140/7127723

Gorchov Negron et al. 2024, Measurement-based carbon intensity of US offshore oil and gas production: https://iopscience.iop.org/article/10.1088/1748-9326/ad489d

*Thank you for this suggestion. We added more introduction and discussion of $CO_2$ emission results to the manuscript and show our $CO_2$ observations explicitly in Figure B2. Also, we added a comparison of carbon intensities for older and newer platforms and as Figure 11. We included some discussion of the facility type emissions and compared with the US offshore emission studies.*

**Comment 2**
Line 63-66. Can the authors back these statements up with numbers? What is the contribution of Africa or at least Nigeria and Angola to the global O&G industry. Why do we think there are major $CH_4$ emissions arising from the production of these resources? Are there onshore studies that have already demonstrated that?

Following lines. How does the oil and gas production in Angola compare with other fields? How much is there? And how much is reinjected? Re-injection is an energy intensive process. Can this explain the $CO_2$ emissions from the deep water facilities?

*We expanded this paragraph to address your questions:*

*"Africa is a significant contributor to the global oil and gas (O&G) industry, accounting for approximately 8% of global crude oil production and 6% of global natural gas production in 2022 (IEA, 2023). Nigeria and Angola are the continent's top producers, together contributing nearly 50% of Africa's oil output. In particular, Angola ranks among the top 20 globally, producing approximately 1.1 million barrels of oil per day in 2022 (OPEC, 2023). Most of Angola's production comes from offshore deep-water fields, which are technically complex and energy-intensive to operate. More particular, the country's offshore oil production is split between older shallow-water platforms closer to the coast and newer deep-water and ultradeep-water fields operated by tethered Floating Production Storage and Offloading (FPSO) vessels that can serve several oil fields at once and therefore have higher production volumes than the shallow-water platforms. Much of the produced natural gas is associated gas from oil fields, and a substantial share is reinjected to maintain reservoir pressure, approximately 65% in recent years according to national reporting (ANPG, 2023). This reinjection process, along with the use of gas turbines for power generation on deep-water facilities, can contribute significantly to $CO_2$ emissions. "*

**Comment 3**
Line 70-72. The text argues that the processing operations for FPSO hydrocarbons occurs at the onshore LNG facility, but the older shallow water facilities do not send hydrocarbons to the LNG facility. Is the correct interpretation that the processing for deep water occurs onshore and the processing for shallow water occurs onsite offshore? This paper finds higher $CH_4$ emissions from shallow water facilities compared to deep water, but is that an artifact of the fact that processing emissions (where the majority of offshore $CH_4$ emissions seem to occur) are just exported to the LNG facility? Are there preliminary processing operations occurring on the FPSOs (like treaters and dehydrators separating the water, gas, and oil) or do even those operations occur at the LNG facility. Can the authors discuss this more?

Was the LNG facility sampled in another flight? If yes, can it be included in this story to round out the processing emissions section of the story?

*We have to correct here: The natural gas is sent to the onshore LNG facility, but the oil is still processed on the FPSO. The gas from the shallow-water facilities is not commercialized and mostly flared. Processing of oil occurs offshore for both types of facilities. We clarified in the text.*

*Actually yes, the LNG facility was sampled twice and shows medium range emissions. We elaborated in the discussion part.*

**Comment 4**

Line 83. There are some additional citations for airborne offshore methane studies that the authors should consider adding.

New regions

Zang, Kunpeng, Gen Zhang, and Juying Wang. "Methane emissions from oil and gas platforms in the Bohai Sea, China." Environmental Pollution 263 (2020): 114486.

- They report concentrations and a regional flux

Gorchov Negron, Alan M., Eric A. Kort, Genevieve Plant, Adam R. Brandt, Yuanlei Chen, Catherine Hausman, and Mackenzie L. Smith. "Measurement-based carbon intensity of US offshore oil and gas production." Environmental Research Letters 19, no. 6 (2024): 064027.

-They add offshore Alaska and California in the US

More Gulf of Mexico studies

Gorchov Negron, Alan M., Eric A. Kort, Stephen A. Conley, and Mackenzie L. Smith. "Airborne assessment of methane emissions from offshore platforms in the US Gulf of Mexico." Environmental science & technology 54, no. 8 (2020): $511_2$-5120.

Ayasse, Alana K., Andrew K. Thorpe, Daniel H. Cusworth, Eric A. Kort, Alan Gorchov Negron, Joseph Heckler, Gregory Asner, and Riley M. Duren. "Methane remote sensing and emission quantification of offshore shallow water oil and gas platforms in the Gulf of Mexico." Environmental Research Letters 17, no. 8 ($202_2$): 084039.

Biener, Kira J., Alan M. Gorchov Negron, Eric A. Kort, Alana K. Ayasse, Yuanlei Chen, Jean-Philippe MacLean, and Jason McKeever. "Temporal variation and persistence of methane emissions from shallow water oil and gas production in the Gulf of Mexico." Environmental Science & Technology 58, no. 11 (2024): 4948-4956.

-They compare with satellite like your study here so may be a good comparison point.

*We added these studies here and in the discussion. Please also see below.*

**Comments on Measurements and Mass Balance:**

**Comment 5**
Figure 1. Since no lat or lon is offered in Figure 1a, can the authors include points of the offshore facilities or show a separate map of facilities? The facilities are shown later in the inventory figures and it would make sense to not cover up the aircraft flights with the facilities, but showing them all together might make for an informative figure of what the campaign looked like. In this map, can the authors include a shallow vs deep water dividing line and the LNG facility?

*We added the facilities and the shallow vs. deep water dividing line to Figure 1.*

**Comment 6**
Line 600 & Figure A2. Can the authors show what $CH_4$ and $CO_2$ look like in the vertical profiles? Since these are your target gases, their level of mixing in the boundary layer is the most important consideration. If these are included in the criteria, would it change your decisions on the height of the PBL or whether the PBL is mixed enough?

*We added $CH_4$ and $CO_2$ to Figure A2. $CH_4$ is well-mixed in the boundary layer, while $CO_2$ shows a slight increase with height. Both trace gases have distinctly higher concentrations in the free troposphere.*

**Comment 7**
Can the authors include a figure (perhaps a multi-panel plot of multiple examples, perhaps in the appendix), showing what the downwind concentrations used for mass balance looked like (something like distance or longitude for the X-axis, and concentration for the Y-axis). Perhaps the authors can include an example that shows what part of the measurements were used for the background. In at least one example, can the authors show what the upwind concentrations looked like as well? The paper states that upwind transects are rare and not used as the background, but were low and therefore the sides of the downwind plumes were sufficient to be used as a background. It would be good to show this.

*We added the requested figure to the appendix. Downwind enhancements and background values are shown. If available the upwind background was added as a horizontal line, representing the average value used for the determination of the upwind background uncertainty.*

**Comment 8**
Line 255. What are the satellite products in the IMEO data portal that provided positive detections in your domain in this study? Who estimated the methane emissions (the satellite data product creator/team?, a team at IMEO?). How was uncertainty assigned to these?

*"MARS draws data from nearly a dozen satellites and space sensors, including the global mapping satellite Sentinel-5P and the high-resolution satellites EnMAP, PRISMA, Sentinel-2, Landsat constellation (from Landsat4 to Landsat9), the EMIT sensor, Sentinel-3, VIIRS sensors, the geostationary satellites GOES and MTG, the newly launched MethaneSAT and Carbon Mapper's Tanager-1. Data is collected daily and analysed by experts with the support of an IMEO artificial intelligence (AI) tool. IMEO experts analyse and validate every detected plume and provide an estimate of emissions with its uncertainty range based on satellite measurements and wind reanalysis data products."* https://methanedata.unep.org/

*We added some of this information to our manuscript. More detailed information can be found on the MARS-website (*https://methanedata.unep.org/methane-alert-response-system*) and given by the MARS team.*

**Comment 9**

Line 578. How does the method create more flux uncertainty with fewer transect flown?

*If there is only one transect, the plume height used for plume mixing height uncertainty is always half of the PBLH, separated into two parts: Ground uncertainty height from the ground to the middle between the ground and plume top uncertainty height from the middle between transect and PBLH up to the PBLH. If there are more than one transects, the two heights decrease because the distance between the individual transects is not considered uncertain. Please see the visual representation below.*

[Figure]

**More Comments:**

**Comment 10**

Figure 3. How many facilities are in each facility group/cluster?

*We included the number of facilities in each group in Figure 4.*

**Comment 11**

Line 299. The text states that the operators reported normal operations during high-emission events and therefore, they must be unaware of their high emissions. I'm not sure this logically flows and it implies that the emissions must be unknown fugitives. Operators can emit large volumes under normal operating conditions when especially they intentionally vent gas. This is one of the driving sources of regular intermittent emissions in the Gulf of Mexico (relevant studies cited elsewhere).

*We agree that large emissions can occur even during what operators consider "normal operations," particularly if intentional venting is part of routine practices. To avoid implying that high emissions necessarily result from unknown fugitives, we revised the sentence to: "Notably, during the high-emission events, the operators reported normal operations and no venting, suggesting they were likely unaware of the emissions. Such emissions are particularly difficult to mitigate, as a lack of awareness prevents timely detection and response." In Angola, intentional venting must be reported in detail to the authorities, and no such reports were filed for these periods. Therefore, we conclude that these emissions were*

*either unintentional or not recognized as venting by the operators. This kind of emission is particularly difficult to mitigate due to the lack of awareness.*

**Comments on Context around Deep vs Shallow water:**

**Comment 12**

Line 355+. The authors highlight how facility condition (e.g. age and type) are more predictive of emissions. Can the authors include photos of the facilities and highlight how deep water and shallow water visibly look different? What do the groups of facilities in shallow water look like? If photos were unfortunately not taken during the campaign, can the authors track down operator or satellite imagery of the facility? The importance of this is emphasized in the following comments. There needs to be more work to characterize what the shallow water infrastructure looks and behave like.

*We did take many pictures from the aircraft. We are sorry, but we cannot include these into the manuscript because of operator anonymity. Nevertheless, we acknowledge the interest in platform type and design. The shallow-water facilities actually do match with the type with high emissions found in the GOM and North Sea regions. We added some more explanation and whenever necessary some discussion and comparison to other regions.*

*"There are two main types of offshore oil and gas facilities in Angola: older shallow-water platforms and newer deep- and ultra-deep-water installations. The shallow-water facilities are typically fixed platforms standing on the seabed. These often form multi-platform complexes, with additional satellite platforms functioning as wellheads. In such cases, the entire complex is considered a single facility. The largest of these includes up to 28 interconnected platforms or wellheads. There are 36 of these older, shallow-water facilities in Angola. In contrast, deep-water operations are conducted from Floating Production, Storage, and Offloading (FPSO) vessels, often converted oil tankers that are moored to the seafloor and connected via flexible pipelines to subsea wellheads. These FPSOs serve as both production and storage units, enabling oil extraction in areas far from shore. Our study includes 21 deep-water facilities."*

**Comment 13**

One major conclusion is that older, low producing shallow water facilities had higher emissions than younger deep water facilities. Research in the Gulf of Mexico also found this trend. This is the second oil and gas basin with (a) clearly distinct shallow water (old) and deep water (new) fields that was also (b) sampled with aircraft. It agrees with the Gulf of Mexico shallow-deep water finding. I suggest the authors consider discussing whether this marks a trend.

*Also, we added a comparison of carbon intensities for older and newer platforms as Figure 11. We included a discussion of the facility type emissions and compared with the US offshore emission studies.*

**Comment 14**

Old shallow water facilities across the globe tend to look similar (based on those visible in google earth). They are composed of a large multiplatform central processing facility

(composed of vents, flares, compressors, etc.) surrounded by smaller satellite production facilities that have almost no infrastructure and basically serve as a well-head. The study on Southeast Asia (Nara), the US Gulf of Mexico studies (Gorchov, Ayasse, Biener), and a satellite Mexican Gulf of Mexico study (see below) all sampled this specific type of infrastructure, and the Southern North Sea paper (Pühl), may have sampled at least one of these facilities. These studies agree in that they found relatively high methane emissions with some identifying highly intermittent emissions from these facilities. The Gulf of Mexico studies also found higher shallow water emissions (due to this class of old facility) than deep water.

In this study, the clusters of facilities in shallow water had high methane emissions that were intermittent, so this raises the question of whether this is the same type of facility as in these other studies.

Do your photos show these facility groups to match this description? If you do not have photos and cannot see the facilities from google earth pro/satellite products, what do the pipelines show? If there are pipelines transporting hydrocarbons to one facility from all of the others, this likely matches this facility class. Are you sure these are groups of separate facilities or are they different parts of one multiplatform facility connected by a cat-walk? If it is a multiplatform facility, then it might also match this facility class.

If yes, I suggest the authors (A) discuss this, contextualize the finding with trends in the literature, and note how previous studies of this facility class also found high intermittent methane emissions. Venting was found to be one of various sources of intermittent high emissions. Do photos show vents on these facilities? This assumes photos were taken during the airborne campaign.

I also suggest the authors (B) consider whether dividing by the count of facilities in the cluster is misleading since they are not independent and treat the cluster as one unit in their denominator for emissions per facility. Please note that the emissions/unit for the Gulf of Mexico calculation treated facility grouping of this specific design as 1 unit (so 1 processing facility with 20 satellite production facilities was considered to be 1 facility). If the cluster of facilities match this type of infrastructure, then make sure the comparison is consistent.

Irakulis-Loitxate, Itziar, Javier Gorroño, Daniel Zavala-Araiza, and Luis Guanter. "Satellites detect a methane ultra-emission event from an offshore platform in the Gulf of Mexico." Environmental Science & Technology Letters 9, no. 6 (2022): 520-525.

*Thanks for this comment. The shallow-water facilities in Angola seem top be of the exact same type as the facilities in Southeast Asia and GOM mentioned above. We have pictures of them and they fit the description. Also, they show the same characteristics and we added this to the text.*

*(A) We added a comparison with the trends in literature in Section 3.5 and also extended Figure 9 with the shallow vs. deep water comparison.*

*(B) We did treat the multiplatform facilities as one facility in our analysis and calculations. This is implicit, as it is not possible to distinguish the emissions from a multicomplex at the distances that we measured the plumes. We clarified in Section 3.1: "The shallow-water facilities are typically fixed platforms standing on the seabed. These often form multi-platform complexes, with additional satellite platforms functioning as wellheads. In such cases, the*

*entire complex is considered a single facility. The largest of these includes up to 28 interconnected platforms or wellheads."*

**Comment 15**

Section 3.4. I suggest you consider if your satellite vs. airborne story is similar to satellite vs. airborne comparisons in…

Biener, Kira J., Alan M. Gorchov Negron, Eric A. Kort, Alana K. Ayasse, Yuanlei Chen, Jean-Philippe MacLean, and Jason McKeever. "Temporal variation and persistence of methane emissions from shallow water oil and gas production in the Gulf of Mexico." Environmental Science & Technology 58, no. 11 (2024): 4948-4956.

*We added a comparison of the two studies in Section 3.4.*

**Comments on Presenting Final Results.**

**Comment 16**

Line 425 & Figure 9. If you are using the emissions/facility reported in Figure 4 from Pühl et al. (2024) for the comparison, please note that the emissions/facility for the Gulf of Mexico are from the 2020 Gulf of Mexico paper and not the 2023 paper.

Gorchov Negron, Alan M., Eric A. Kort, Stephen A. Conley, and Mackenzie L. Smith. "Airborne assessment of methane emissions from offshore platforms in the US Gulf of Mexico." Environmental science & technology 54, no. 8 (2020): 5112-5120.

The 2023 paper is more extensive and can be used to make a complete ratio for shallow water and deep water. The 2020 paper sampled a fraction of GOM facilities so the balance of shallow vs deep water will change that ratio.

*We changed the numbers and references accordingly.*

**Comment 17**

Figure 9. Since the deep water and shallow water regions have such different infrastructure and emissions, I suggest also making a separate comparison with a deep water emissions/facility and a shallow water emissions/facility. Perhaps it may be interesting to compare an emissions/facility for the deep water Gulf of Mexico and shallow water Gulf of Mexico to see if they match.

*We added the average emissions for Angolan old and new facilities and also compared to the Gulf of Mexico studies. We did not find emission factors for individual facilities of GOM in the newer publications.*

**Comment 18**

Line 440. How is error calculated for total emissions with the sum of the mean?

*We clarified: "The total Angolan offshore emissions of CH4 and CO2 derived by airborne mass balance during MTGA is calculated as the sum of the emissions from all facilities. The total error is the sum of all errors."*

**Comment 19**

If the deep water and shallow water systems are different, can you estimate a separate carbon intensity for each in addition to the combined carbon intensity? Are the deep water facilities dominated by $CO_2$ and the shallow water facilities dominated by CH4? How does this compare to the deep water and shallow water carbon intensity and GHG breakdowns in at least the Gulf of Mexico?

*We included this comparison in the new Figure 10 and discussion thereof.*

**Comment 20**

Line 495. Do the authors include reinjected gas in the production denominator for carbon intensity and methane intensity? Be careful with this. I'm of the mind that you should not include it as what policy makers care about is how much emissions come from the marketed oil and gas.

*We agree and did not include the reinjected gas in the denominator of the carbon and methane intensities. We clarified this in Section 3.6.*

**Comment 21**

Line 645-Data accessibility. Will the authors share their flux data too? This data could be useful for meta-analyses. I know there is caution about sharing operator information. Would publishing flux data with just lat and lon get around that?

*We are very sorry, but we are not able to share the flux data in connection with lat/lon, because of operator integrity.*

**Technical Comments.**

**Comment 22**

Line 37-The wording is a little redundant with second most important GHG point in line 34.

*Fixed.*

**Comment 23**

Line 159-vertical or horizontal transect? Is the correct reading… "Summing up the fluxes from all *horizontal* transects, *vertically*,…"? Currently it reads as summing up vertical transects.

*You are correct. We changed this.*

**Comment 24**

Figure 2: Arrows are hard to see.

*We changed the color of the arrows to black for better visibility.*

**Comment 25**
Figure 4: This is a log plot so hard to see if daily variability is in fact generally low or not.

*Day-to-day variability is also discussed in the text of Section 3.1.*

**References:**

ANPG: Boletim Informativo da Produção – Dezembro 2022, ANPG, https://www.anpg.co.ao, 2023.
IEA: World Energy Outlook 2023, International Energy Agency, https://www.iea.org/reports/world-energy-outlook-2023, 2023.
OPEC: OPEC Annual Statistical Bulletin 2023, Organization of the Petroleum Exporting Countries,, https://asb.opec.org, 2023.